# INI1/SMARCB1 Rpt1 domain mimics TAR RNA in binding to integrase to facilitate HIV-1 replication

Updesh Dixit[1,6], Savita Bhutoria[1,6], Xuhong Wu[1], Liming Qiu [2], Menachem Spira[1], Sheeba Mathew[1], Richard Harris[3], Lucas J. Adams[4], Sean Cahill [3], Rajiv Pathak[1], P. Rajesh Kumar[3], Minh Nguyen[1], Seetharama A. Acharya[5], Michael Brenowitz [3], Steven C. Almo[3], Xiaoqin Zou [2], Alasdair C. Steven [4], David Cowburn[3], Mark Girvin[3] & Ganjam V. Kalpana [1✉]

INI1/SMARCB1 binds to HIV-1 integrase (IN) through its Rpt1 domain and exhibits multi-faceted role in HIV-1 replication. Determining the NMR structure of INI1-Rpt1 and modeling its interaction with the IN-C-terminal domain (IN-CTD) reveal that INI1-Rpt1/IN-CTD interface residues overlap with those required for IN/RNA interaction. Mutational analyses validate our model and indicate that the same IN residues are involved in both INI1 and RNA binding. INI1-Rpt1 and TAR RNA compete with each other for IN binding with similar $IC_{50}$ values. INI1-interaction-defective IN mutant viruses are impaired for incorporation of INI1 into virions and for particle morphogenesis. Computational modeling of IN-CTD/TAR complex indicates that the TAR interface phosphates overlap with negatively charged surface residues of INI1-Rpt1 in three-dimensional space, suggesting that INI1-Rpt1 domain structurally mimics TAR. This possible mimicry between INI1-Rpt1 and TAR explains the mechanism by which INI1/SMARCB1 influences HIV-1 late events and suggests additional strategies to inhibit HIV-1 replication.

[1] Department of Genetics, Albert Einstein College of Medicine, New York City, NY, USA. [2] Dalton Cardiovascular Research Center, Department of Physics and Astronomy, Department of Biochemistry, and Institute for Data Science and Informatics, University of Missouri, Columbia, MO, USA. [3] Department of Biochemistry, Albert Einstein College of Medicine, New York City, NY, USA. [4] Laboratory of Structural Biology Research, National Institute of Arthritis and Musculoskeletal and Skin Diseases, National Institutes of Health, Bethesda, MD, USA. [5] Department of Anatomy & Structural Biology, Albert Einstein College of Medicine, New York, NY, USA. [6] These authors contributed equally: Updesh Dixit, Savita Bhutoria. ✉email: ganjam.kalpana@einsteinmed.org

INI1/SMARCB1/hSNF5/BAF47 is an invariant component of the SWI/SNF chromatin remodeling complex, involved in a multitude of cellular functions, including transcription, cell cycle regulation, development, and tumor suppression[1,2]. INI1 and SWI/SNF are frequently mutated in cancers[1,2]. INI1 was first identified as a binding partner for HIV-1 integrase (IN)[3], and studies suggest that it is required at multiple stages of HIV-1 replication including integration, HIV-1 transcription, post transcriptional Gag RNA and protein stability, virus assembly, and particle production[4–14]. Interestingly, HIV-1 IN also influences multiple stages of viral replication, including integration and particle morphogenesis[15–18].

INI1/SMARCB1 interacts with various viral and cellular proteins[8,19–23] via its two highly conserved imperfect repeat domains, Rpt1(aa 183–248) and Rpt2(aa 259–319), connected by a linker region (aa 249–258) (Fig. 1a)[24]. Rpt1 (but not Rpt2) is necessary and sufficient for binding HIV-1 IN[24]. INI1 is selectively incorporated into HIV-1, but not other lentiviral or retroviral particles[11]. Furthermore, an INI1 fragment termed S6 (aa 183–294) harboring the Rpt1 domain, linker region and a part of Rpt2, trans-dominantly inhibits HIV-1 particle production by

binding to IN within GagPol[10]. These studies indicate that INI1 is required for HIV-1 late events. INI1 influences integration in vitro depending on IN-INI1 stoichiometry, and the addition of SWI/SNF complex enhances integration into nucleosomal targets[3,5,13]. How INI1 binds to IN and influences multiple stages of HIV-1 replication is not clearly understood.

Here, we provide the solution structure of the conserved Rpt1+ linker domain (INI1$_{183-265}$) of INI1 (PDB ID: 6AX5) and molecular docking of the IN-CTD/INI1-Rpt1 complex. These studies indicate that the IN residues at the IN/INI1 interface are the same as those needed for the interaction of IN with the TAR region of the HIV-1 genome[18]. IN/RNA interaction is necessary for particle morphogenesis[15,18]. Interestingly, IN mutants defective for binding to INI1 also affect particle morphogenesis[7]. Our current analysis indicates that there is a remarkable similarity between INI1-Rpt1 and TAR RNA with regard to their binding to IN-CTD. Comparison of the negative charges on the electrostatic surfaces of the INI1-Rpt1 and the TAR RNA NMR structures, and modeling of the IN-CTD/TAR RNA complex by molecular docking and its comparison to IN-CTD/INI1-Rpt1 complex reveals that INI1-Rpt1 may structurally mimic TAR RNA. The

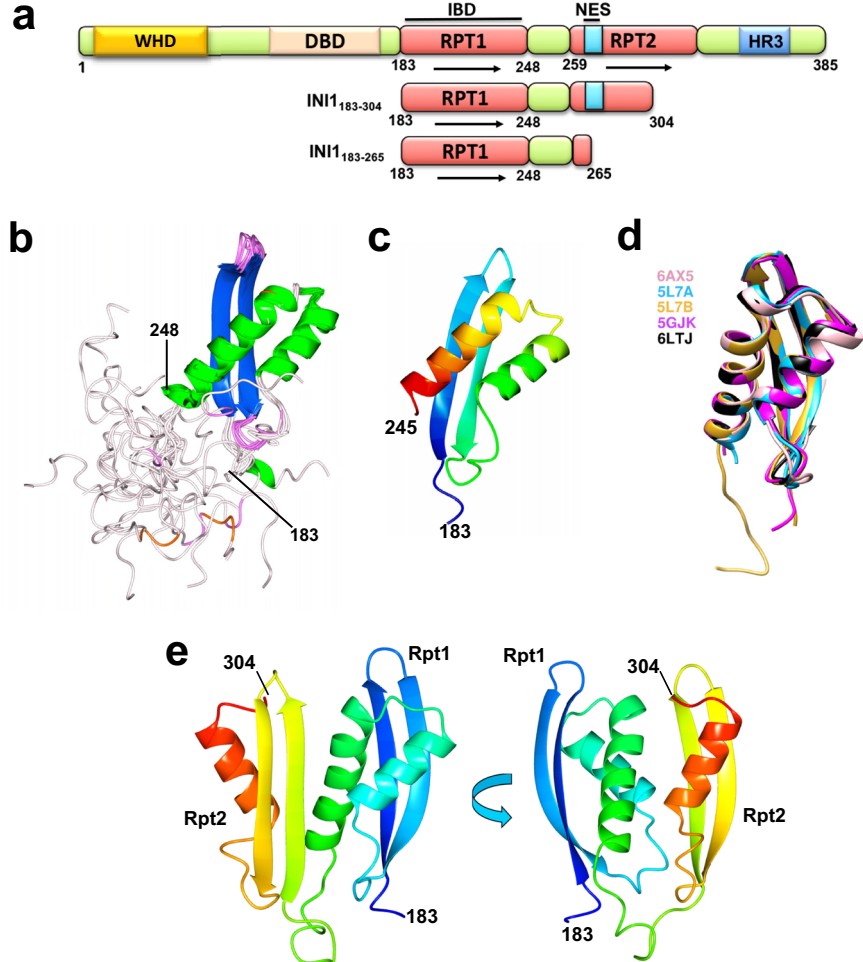

**Fig. 1 NMR structure of Rpt1 + linker domains of INI1 and molecular modeling of IN binding Rpt1 + linker + Rpt2 fragment. a** Schematic representation of various domains of INI1 and the inhibitory fragment S6 (WHD (in yellow) = Winged Helix DNA binding domain, DBD (in cream) = DNA binding domain; RPT (in red) = Repeat; NES (in turquoise) = Nuclear export signal; HR3 (in blue) = homology region 3; arrows represent repeats). **b** Superposition of the residues 183-265 of the 20 lowest-energy NMR structures of the INI1 Rpt1+linker fragment. Note the disordered nature of the linker region (aa 250-265, shown in pale pink). The helices are in green and beta-sheet in blue. **c** A ribbon diagram (in rainbow colors) of a lowest energy representative structure of Rpt1 (aa183-245). **d** Superimposition of C-alpha atoms of five different structures (6AX5 in pink, 57LA in turquoise, 5L7B in yellow, 5GJK in magenta, 6LTJ in black) showing alignment of the Rpt1 region (aa 183-248); **e** Ribbon diagram (in rainbow color) of a representative structure of INI1$_{183-304}$ modeled using Robetta based on the NMR structure 6AX5. PDB file of this model is included in the Supplementary Data 1.

structural mimicry between INI1-Rpt1 and TAR RNA explains the multifaceted role of INI/SMARCB1 during HIV-1 replication in vivo and provides mechanistic insights into INI1–IN interactions.

## Results

**NMR structure of INI1$_{183-265}$ and modeling INI1$_{183-304}$.** We selected INI1$_{183-265}$ for NMR study after screening several overlapping fragments harboring the Rpt1 domain, as this fragment exhibited good solution property (Fig. 1a and Supplementary Figs. 1 and 2). The uniformly $^{13}$C,$^{15}$N-labeled INI1$_{183-265}$ fragment was subjected to NMR analysis. The assigned $^{1}$H$^{15}$N HSQC spectrum for non-deuterated INI1$_{185-265}$ is shown in supplementary Fig. 3b. Unlike the multimeric full-length protein, the INI1$_{183-265}$ domain is monomeric in solution as judged by both NMR self-diffusion and analytical ultracentrifugation measurements (Supplementary Fig. 3b). The INI1$_{183-265}$ fragment consists of the Rpt1 domain (aa183–248) and the linker region (aa 249–265) between Rpt1 and Rpt2. The NMR structure indicated the presence of a well-ordered Rpt1 domain-containing ββαα topology and a disordered linker segment [Fig. 1 b, c, Supplementary Table 1, and PDB ID: 6AX5]. Superimposition of our structure (6AX5) to other existing NMR, X-ray crystal, and cryoEM structures [5L7A, 5L7B, 5GJK, and 6LTJ] indicated that the ββαα core region of 6AX5 perfectly aligns with the other structures with an RMSD (root-mean-square deviation) of ~1.0 Å (Fig. 1d and Supplementary Table 2).

While Rpt1(INI1$_{183-245}$) is sufficient for IN binding, a longer fragment S6(INI1$_{183-294}$) harboring Rpt1+linker+partRpt2, shows stronger binding[24] and acts as a dominant-negative inhibitor of HIV-1[10]. Consistent with this, the fragments INI1$_{183-304}$ and INI1$_{166-304}$ but not INI1$_{183-265}$ interacted strongly with full-length IN and central core (IN-CCD, aa 50–200) and C-terminal (IN-CTD, aa 201-288) domains in vitro (Supplementary Fig. 4 a-f). To determine the contribution of INI1-Rpt2 for IN binding, we modelled INI1$_{183-319}$ fragment containing Rpt1-linker-Rpt2 based on 6AX5 structure using Robetta[25]. This modeling simulation yielded five best clusters, each possessing well-ordered Rpt1 and Rpt2 domains. The Rpt2 region was topologically identical to Rpt1 and the five clusters differed from each other only in the linker region (Supplementary Fig. 5a and Supplementary Data 1). We also modeled the structure of the strong binding INI1$_{183-304}$ fragment by using Robetta[25], which showed that Rpt1 and Rpt2 domains were symmetrically arranged separated by the linker region (Fig. 1e and Supplementary Data 1). Further ab initio modeling of linker region in INI1$_{183-304}$ exhibited significantly greater order than that observed in the NMR structure that lacked the Rpt2 (Fig. 1e). This modeled structure was validated using various tools including Ramachandran plot, PROCHECK version 3.5[26] WHATIF ver. 8.0[27], VERIFY3D v3.1[28], and PROSA 2003[29] (Supplementary Fig. 5b–d), which strongly supported the features of the model. The highest scoring modeled INI1$_{183-304}$ structure was used to study its interaction with IN. Analysis of the Rpt1 portion of this model with the existing structures exhibited RMSD of ~1.0 Å (Supplementary Table 3), and Rpt2 portion of this model exhibited RMSD of 1.25 Å with the cryoEM structure[30] (PDBID:6LTJ).

**Interaction of INI1 and IN is mediated by extensive hydrophobic and complementary ionic interactions.** While INI1$_{183-304}$ fragment strongly binds to IN-CCD and IN-CTD domains (Supplementary Fig. 4), the INI1$_{183-304}$/IN-CTD complexes were large and hence were not amenable for NMR studies. Therefore, we computationally docked the INI1$_{183-304}$ structure with the NMR structure of IN-CTD [(1QMC)][31,32] using HADDOCK, without

interaction restraints (Fig. 2 and Supplementary Data 1). The docked complex with the lowest (best) energy HADDOCK score of -184.3 ± 21.5 indicated that the Rpt1 portion within the INI1$_{183-304}$ fragment has the potential to directly interact with IN-CTD (Fig. 2a). The exposed negatively charged Rpt1 residues of INI1$_{183-304}$ contacted the basic residues of IN-CTD, and the interaction of hydrophobic residues of the two proteins resulted in a hydrophobic core within the complex (Fig. 2b,c and f). Upon complex formation, ~642 Å$^2$ of solvent-accessible surface was buried, which is similar in size to buried surface area in the comparable SWIRM/ INI1 Rpt1 complex [699.1 Å$^2$][33].

This docking study suggested that the INI1-Rpt1/IN-CTD interaction was mediated by both hydrophobic and specific ionic interactions (Fig. 2d–f). In the model of the complex, acidic residues D225 and D224 from α1 of INI1-Rpt1 made polar contacts with basic IN-CTD residues R228 (in strand β1), K264, and R263 (in the loop between β4 and β5 strands) (Fig. 2d). Additional ionic interactions were noted between INI1$_{183-304}$ E210 (in the loop region between β2 and α1) and IN-CTD residues R262 (in the loop between β4 and β5 strands) and K244 (in the loop between β2 and β3 strands) (Fig. 2f). The IN-CTD W235 residue formed the center of the Rpt1/CTD interface, surrounded by a shallow hydrophobic patch/cage formed by F204, L226, L222, I221, and F228 residues from α1 of Rpt1 (Fig. 2e, f). This hydrophobic channel is surrounded by the ionic interactions formed between INI1-Rpt1 residues D225, D224 with IN-CTD residues R228, K264, and R263 at one end; between E210 of INI1 with K244 and R262 of IN-CTD, and between E184 of INI1 and R269 of IN-CTD residues at the other end. The validity of this model was confirmed by biochemical studies of interface residue mutants as follows.

**Biochemical interaction studies validate the INI1$_{183-304}$/CTD model.** Previous reverse yeast-two hybrid genetic screening using a random mutation library identified D225G, T214A, and D227G (termed E3, E4, and E10) mutations in S6(INI1$_{183-294}$) that disrupted its ability to interact with integrase and inhibit HIV-1 particle production (Fig. 3a)[10]. D225G and T214A mutants were most defective for binding and inhibition, while the D227G mutant was less defective[10]. In our model, while D225 residue is at one end of α1 helix and makes ionic interactions with IN R228 and K264, T214 is at the other end of the INI1-Rpt1 α1 helix, facing the binding interface (Fig. 2d and Supplementary Fig. 6a). Therefore, mutating these residues would disrupt the IN/INI1 interaction or destabilize the α1 helix. Interestingly, INI1-Rpt1 D227 faces away from the binding interface and substituting this residue should cause minimal disruption of the interaction (Fig. 2d). To test these predictions, we determined the interactions of GST-fusions of full-length IN, CCD, and IN-CTD with INI1$_{183-304}$ wild type and D225G, D227G, T214A mutants (Fig. 3b). We found that INI1$_{183-304}$ mutants D225G and T214A were highly defective and D227G mutant was least defective, for binding to IN and IN-CTD, consistent with the prediction of the model (Fig. 3b).

Our model also predicts that W235 residue is nestled in a hydrophobic cage, and substituting this residue with a charged but not with another aromatic residue would disrupt IN/INI1 interactions (Fig. 2e, f). Interestingly, previous reports have indicated that W235E and W235K, but not W235F substitution mutations selectively inhibit integration activity in vivo but not in vitro[34–36]. The reason for the differential effects of these mutants in vivo are not well understood. We carried out an in vitro GST-pull down assay to determine if the phenotypes observed for W235 mutants correlate with their ability to interact with INI1. IN-W235E mutant was defective for interaction with

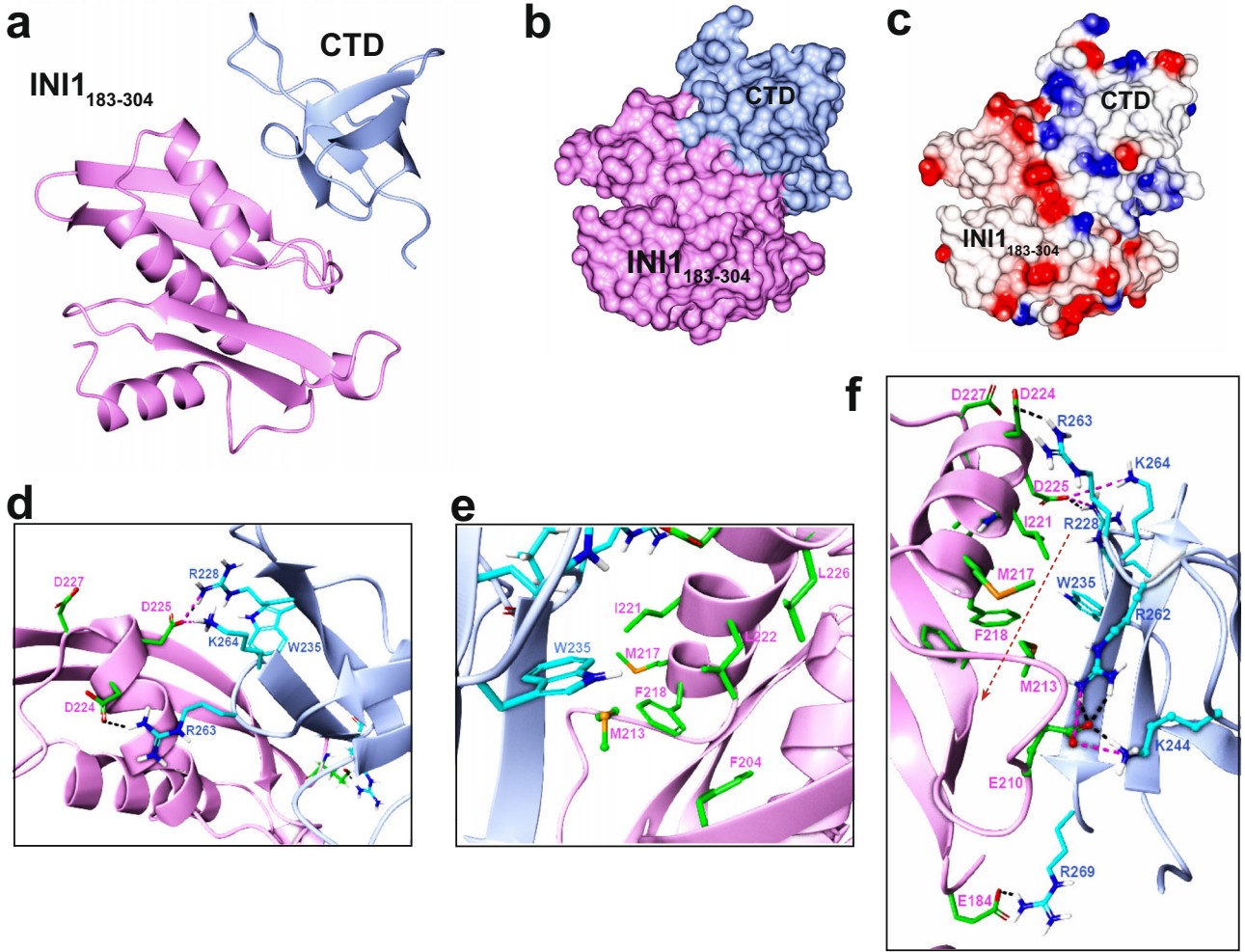

**Fig. 2 Molecular docking of IN-CTD with INI1$_{183-304}$. a** Ribbon diagram of bound complex of INI1$_{183-304}$/CTD model obtained from HADDOCK. **b** Surface structure of INI1$_{183-304}$/CTD modeled complex. **c** Electrostatic surface; negatively charged (red), positively charged (blue) and hydrophobic (white). **d–e** Exploded views of the interface residues, displaying the ionic interactions (**d**), and hydrophobic non polar interactions (**e**). **f** Representation of the regions between INI1$_{183-304}$/CTD showing a hydrophobic tunnel enclosed by ionic bonds at the two ends. In all panels, INI1$_{183-304}$ and its residues are shown in pink, and IN-CTD and its residues are shown in blue. Orange dotted line with arrow represents the hydrophobic tunnel. PDB file of this docked model is included in Supplementary Data 1.

GST-INI1 (Fig. 3c). To determine the specificity, we also tested the interaction of IN-W235E mutant with GST-fusions of other binding partners of IN, namely LEDGF, Gemin2 and SAP18. Our data indicated that while wild type His$_6$-IN binds to all four proteins, the His$_6$-IN-W235E mutant was highly defective for binding to GST-INI1 (<5% binding), partially defective for binding to GST-SAP18 (~20% binding) and was not significantly defective for binding to GST-fusions of LEDGF and GEMIN2 (Fig. 3c). These studies suggested that W235 residue is specifically important for interaction of IN with INI1. We next tested the ability of W235E, W235K, and W235F substitutions to interact with GST-IN. We found that while the W235F mutant retained the interaction with INI1 as strongly as the wild type, W235K and W235E mutants showed reduced interactions in vitro (Supplementary Fig. 6b). These results were also validated by Alpha proximity assay[37,38] and co-immunoprecipitations as described below (Figs. 4 and 5).

**Similarity between the INI1$_{183-304}$ and TAR RNA for binding to IN.** During these studies, we unexpectedly noted that some of the critical residues at the interface of the IN-CTD/INI1$_{183-304}$

complex, such as K264, R269, were the same residues shown to be important for the interaction of IN with HIV-1 genomic RNA[15,18] (Fig. 2d, f). These mutations affect binding of the TAR region of HIV-1 genomic RNA to IN and lead to defective particle morphogenesis[18]. Our previous studies have indicated that INI1-interaction-defective-(IID) IN mutants in the core domain also lead to defects in particle morphogenesis[7]. Based on these observations, we hypothesized that INI1 and TAR RNA could bind to the same residues of IN-CTD and may have overlapping functions in particle morphogenesis. To test this hypothesis, we determined if: (i) INI1$_{183-304}$ and TAR compete with each other for binding to IN in vitro; (ii) IN-CTD mutations affect the binding of INI1$_{183-304}$ and TAR to the same extent; (iii) INI1 can compete with IN binding to TAR in vivo; and (iv) INI1-interaction defective W235E and R228A mutant viruses form morphologically defective particles and are defective for incorporation of INI1 into the virions.

We established a quantitative protein-protein interaction Alpha proximity assay[37,38] to assess the interaction of IN/INI1 and IN-CTD/INI1$_{183-304}$, the results of which showed low nM $K_D$ for binding (Fig. 4a and Supplementary Fig. 7a). While IN-CTD/ INI1$_{183-304}$ interactions were unaffected by salt, IN-CTD/TAR

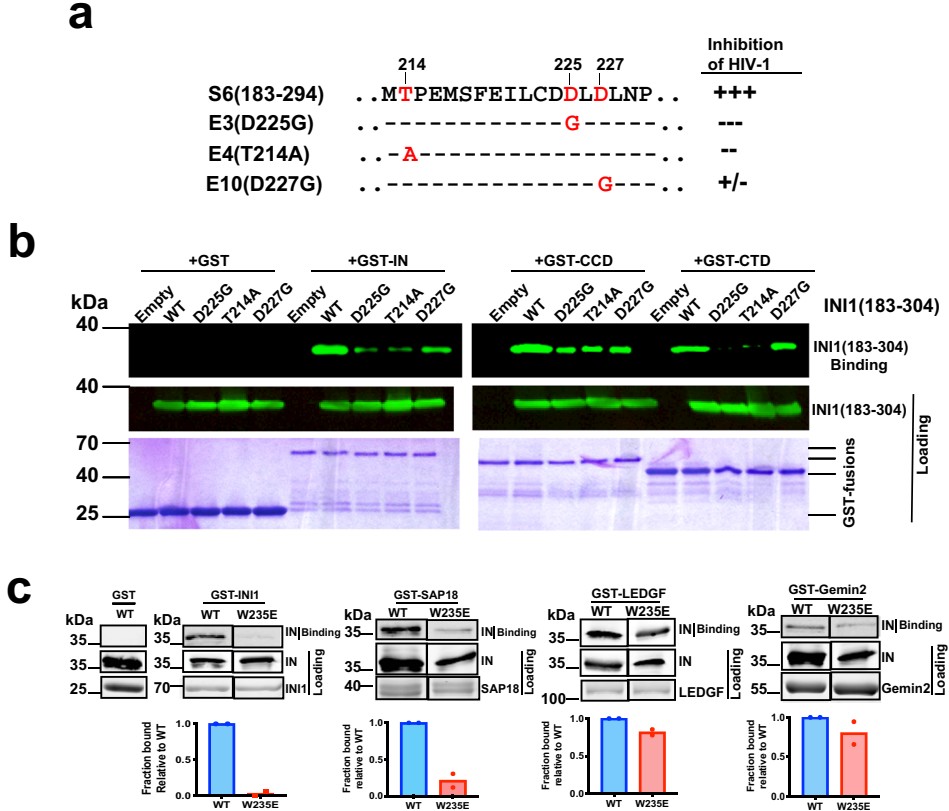

**Fig. 3 In vitro binding studies to validate the interacting interface residues predicted in the CTD/INI1$_{183-304}$ complex. a** Sequence of a portion of S6/Rpt1 fragment of INI1 and the IN-interaction-defective substitution mutations (E3, E4, and E10) identified in a random genetic, reverse yeast two-hybrid screen and their effect on S6-mediated inhibition of HIV-1 particle production. **b** GST-pull down assay to demonstrate the binding of INI1$_{183-304}$ and its mutants with IN, CCD and CTD. Representative images from one out of three experiment is shown. Top panel represents bound proteins and the bottom two panels represent the loading control. Top two panels represent the Western blot using α-BAF47 antibodies to detect 6His-SUMO-INI1$_{183-304}$. Bottom panel represents the Coomassie-stained gel of GST-fusion proteins. **c** GST-pull down assay to determine the interaction of His$_6$-IN(WT), His$_6$-IN(W235E) mutant with GST-INI1, GST-SAP18, GST-LEDGF, and GST-Gemin2. Representative images from one out of three experiment is shown. Top panel represents the bound proteins and the two panels below the top represent the loading controls. Non-adjacent lanes from the same gel are spliced together for the figure and uncropped gels are provided in the source data. Graphs at the bottom represent quantitation of the bound proteins expressed as fraction bound after normalizing to the loading control. The graphs represent the mean of two independent experiments, WT is Wild type IN (shown in blue) and W235E is shown in red.

interaction was sensitive to high NaCl (500 nM), consistent with the previous results[18] (Fig. 4b, c).

First, we used Alpha assay to determine if there is competition between TAR RNA and INI1$_{183-304}$ for binding to GST-CTD. Constant amounts of either Biotinylated-TAR or His$_6$-SUMO-INI1$_{183-304}$ were incubated with GST-CTD, in the presence of increasing concentration of a third molecule (either INI1$_{183-304}$ or TAR RNA) under low salt conditions[37,38]. We found that TAR RNA and INI1$_{183-304}$ competed with each other for binding to IN-CTD with similar IC$_{50}$ value (≈ 5 nM, Fig. 4d and e). The competition was specific, as another fragment of HIV-1 RNA (nts 237-279), containing a similar stem and loop content did not significantly inhibit the interaction between INI1$_{183-304}$ and IN-CTD under these conditions (Supplementary Fig. 7b). These results indicated that TAR and INI1$_{183-304}$ compete with each other to bind to IN-CTD.

Second, we tested to determine if IN-CTD shows the same profile of interactions with both INI1$_{183-304}$ and TAR RNA. We tested the effect of a panel of IN-CTD substitution mutants from the IN-CTD/INI1$_{183-304}$ interface, R228A, K244A, W235E, W235K, W235F, W235A and the two mutants K264A/K266A and R269A/K273A that have been shown to block IN/TAR binding[18] for their ability to interact with INI1$_{183-304}$ and TAR RNA at low salt

conditions using Alpha assay. Our results indicated that all the mutant IN-CTD proteins tested, except for W235F, were defective for binding to INI1$_{183-304}$ consistent with our model (Fig. 4f, g). Interestingly, the same panel of mutants (except for W235F) were also defective for binding to TAR to the same extent (Fig. 4h). W235F IN-CTD mutant was not defective for binding to either TAR-RNA or INI1$_{183-304}$ whereas other mutants were defective for binding to both, suggesting that INI1$_{183-304}$ and TAR RNA recognize the same residues for binding.

Third, to determine if the in vitro results can be recapitulated in vivo, we carried out co-immunoprecipitation (co-IP) and RNA-co-immunoprecipitation (RNA-IP) to test the interactions of full-length IN with INI1 and HIV-1 RNA in vivo. HA-INI1 and YFP-IN or YFP-IN mutants (W235E, W235F, W235A, R228A, R269A/K273A) were co-transfected into 293 T cells and immuno-precipitated using α-HA antibodies. All the mutants of IN and INI1 were expressed at comparable levels in cells as shown by the input control (Fig. 5a, lower two panels). α-HA antibodies immunoprecipitated equal amounts of INI1 in all the samples (Fig. 5a, second panel from the top). INI1 was able to co-immunoprecipitate WT IN and IN(W235F) (Fig. 5a, lanes 1 and 2, upper panel) but not the other IN mutants. Control samples where either INI1 or IN were missing or when isotype IgG antibody was

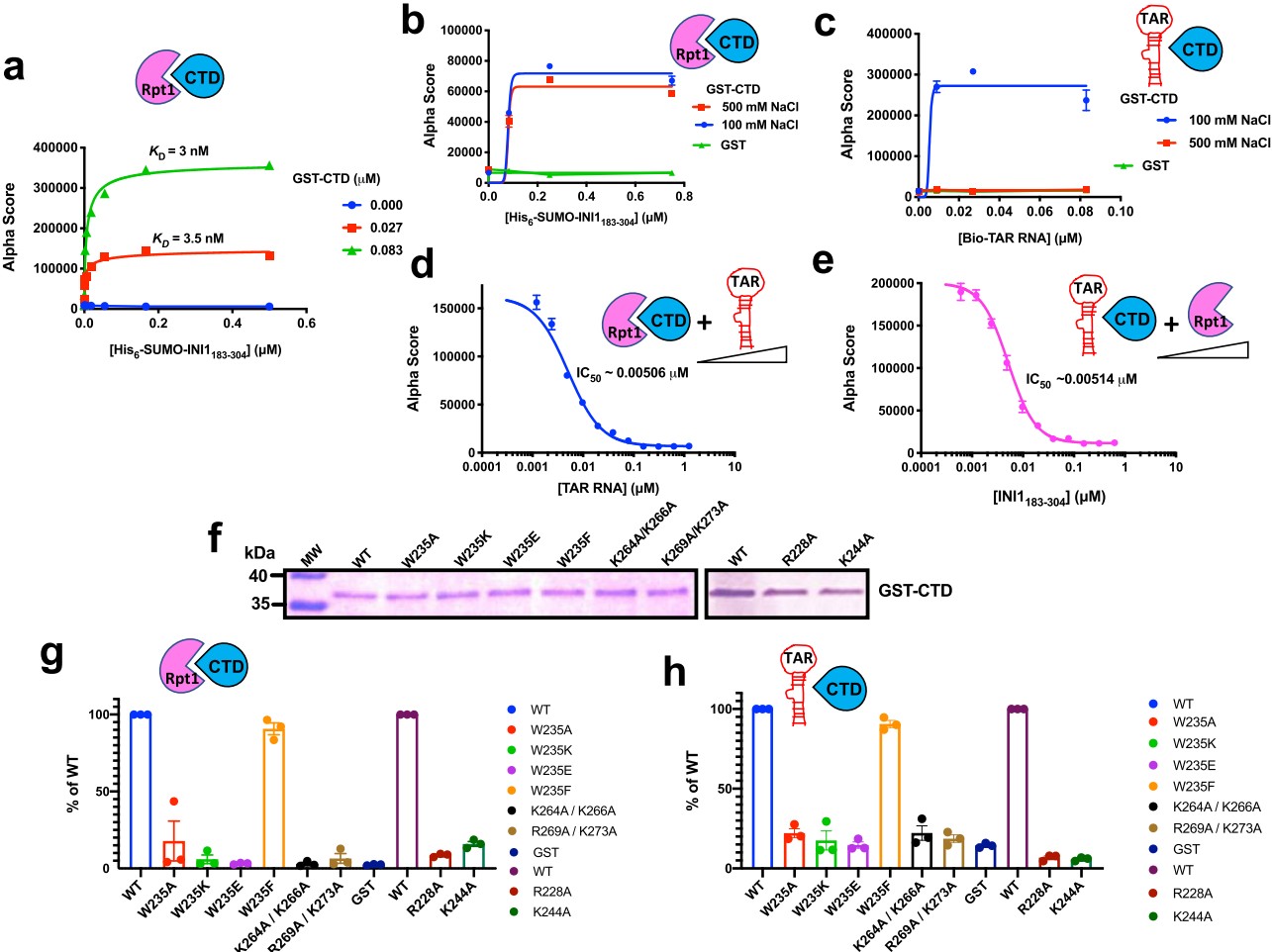

**Fig. 4 Quantitative Alpha protein-protein interaction assay to determine the interaction of IN, CTD, and the mutants with INI1 and INI1_{183-304}.**
**a** Interaction of GST-CTD with His_6-SUMO-INI1_{183-304}. A titration curve was generated with increasing concentrations of His_6-SUMO-INI1_{183-304} with two different fixed concentration of GST-CTD and the interactions were detected as Alpha Score. The $K_D$ values were determined by nonlinear regression analysis using specific binding with Hill slope analysis in GraphPad Prism. Data from one representative experiment is depicted. **b**, **c** Effect of salt on the interactions of GST-CTD either with His_6-SUMO-INI1_{183-304} or biotinylated(Bio)-TAR RNA ($n = 3$ independent experiments). The interaction was tested using fixed concentrations of GST-CTD (0.75 μM) and increasing concentrations of His_6-SUMO-INI1_{183-304} (or Bio-TAR RNA) in two different NaCl conditions (100 and 500 mM); **d**, **e** Inhibition of GST-CTD interaction with His_6-SUMO-INI1_{183-304} or Bio-TAR RNA ($n = 6$ independent experiments). Interactions were set up between GST-CTD (0.186 μM) with His_6-SUMO-INI1_{183-304} (0.094 μM), or GST-CTD (0.03 μM) with Bio-TAR RNA (0.1 μM) and increasing concentration of the third component (indicated in the X-axis) was added to the reaction. The IC_{50} values were determined by fitting the data to a four-parameter dose-response curve using GraphPad prism. **f**–**h** Interaction of GST-CTD and its substitution mutants with INI1_{183-304} and TAR RNA ($n = 3$ independent experiments). Representative Coomassie gel (from one out of three independent experiments) showing equal loading of the wild type and mutant proteins for the binding assays (**f**) and uncropped gels are provided in the source data. Interaction of GST-CTD and mutants with INI1_{183-304} (**g**), and with biotinylated-TAR RNA (**h**). The graphs represent the % of the interaction of mutants as compared to that of wild type (WT) IN set at 100%. For both panels (**g**) and (**h**), WT and mutants are represented in different colors as indicated in the key provided next to the bar graphs. In all panels, except in **a**, graphs represent Mean ± SEM. In all panels the pink cartoon Rpt1 represents INI1_{183-304}, blue cartoon CTD, IN-CTD, and red stem-loop, TAR RNA.

used, showed no co-immunoprecipitation (Fig. 5a, lanes 7–9). These results establish that the IN residues identified at the interface are important for full-length IN-INI1 interaction in vivo.

We next determined if INI1 and TAR RNA competed with each other for binding to IN in vivo by RNA-co-IP in MON (INI1^{−/−}) cells[9]. TAR RNA was expressed by transfecting pLTR-luc and pCMV-Tat in the presence or absence of YFP-IN and HA-INI1. Lysates of transfected MON cells were treated with DNase I to remove residual DNA and subjected to IP by using α-GFP antibodies to pull down YFP-IN and associated complexes. Protein and RNA were separated from the immune complexes, RNA was subjected to RT-PCR to detect viral RNA, and proteins were subjected to Western blot analysis to detect the presence of

YFP-IN and HA-INI1 using α-GFP and α-HA antibodies. The results indicated that while IgG did not co-immunoprecipitate either INI1 or TAR-RNA (Fig. 5b, lane 1), α-GFP antibodies were able to pull down YFP-IN, which in turn, was able to RNA-co-IP TAR RNA in the absence of INI1 (Fig. 5b, lane 2). The presence of HA-INI1 led to co-immunoprecipitation of both INI1 and TAR RNA by IN (Fig. 5b, lane 4). To determine if INI1 and TAR RNA competed with each other for binding to IN, we transfected increasing concentrations of INI1 and a constant amount of IN. We found that increasing INI1 decreased the amount of bound TAR RNA but increased the amount of bound INI1 (Fig. 5b lanes 5-7, top three panels). We also ran all the input controls, which showed that there was a uniform loading of IN and RNA (Fig. 5b, bottom three panels).

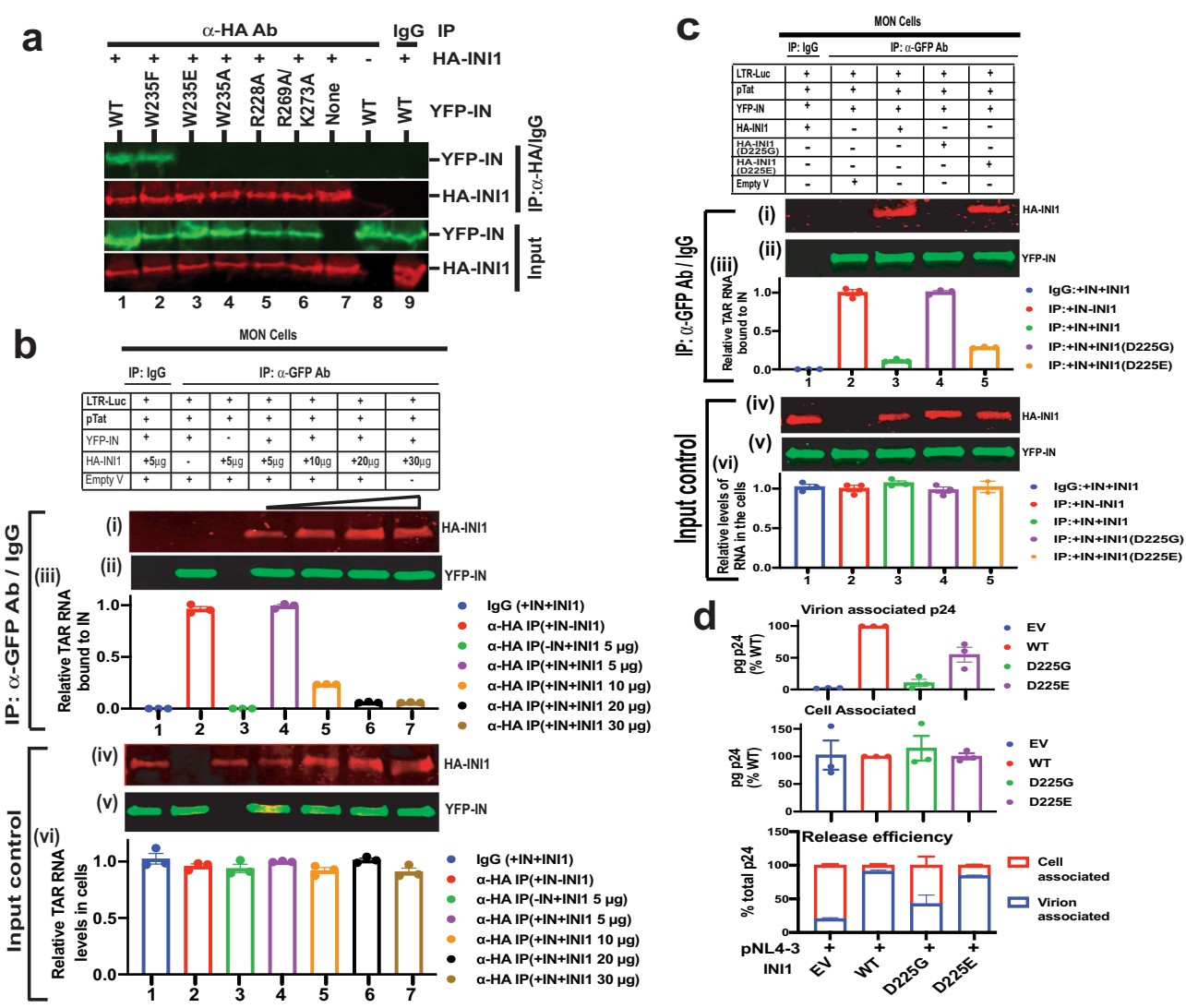

**Fig. 5 INI1 competes with TAR RNA for binding to IN in vivo and facilitates particle production. a** Co-immunoprecipitation of INI1 with IN and mutants in vivo. MON cells were transfected with YFP-IN/IN mutants and HA-INI1, and then subjected to co-immunoprecipitation using α-HA antibodies. Representative images from one out of three independent experiments is shown. The top two panels illustrate results of co-immunoprecipitation using α-HA antibodies. The lower two panels represent the input control. Lanes 7-9 represent controls, use of isotype IgG antibody (lane 9) or lack of INI1 or IN (lanes 7-8). **b** RNA-co-IP analysis to determine the competition of INI1 and TAR RNA for binding to IN. The gel images are from one out of three representative experiments. The top three panels (i)–(iii) represent the RNA and proteins present in the immune complexes and the bottom three panels (iv)–(vi) represent proteins and RNA in the input controls. Panels (iii) and (vi), graphic representation of relative amounts of TAR RNA bound compared to control, as determined by qRT-PCR ($n = 3$ independent experiments), normalized to control (lane 2). Immunoprecipitation was carried out using isotype IgG antibody (lane 1) or α-GFP antibodies to pull down YFP-IN (lanes 2-7). Lane 3 represents negative control without YFP-IN. Lanes # 4-7 represent RNA-co-IP in the presence of increasing HA-INI1. **c** IN-interaction-defective INI1 mutant do not compete with TAR in vivo. The gel images are from one out of three representative experiments. Panels (i)–(vi) are as in **b**. Lanes represent results of RNA-co-IP of YFP-IN with TAR RNA in the presence of WT INI1 (lane 3), INI1(D225G) (lane 4), and INI1(D225E) (lane 5). **d** INI1 binding to IN is necessary for particle production. The top panel represents virus-associated and the middle panel represents the cell-associated p24, expressed as % of wild type. The bottom panel represents release efficiency as a fraction of viral- and cell-associated p24 of each mutant, as compared to wild type, expressed in %. EV = Empty vector; WT = wild type. The bars represent the average of three independent experiments (Mean ± SEM). The bars are color-coded as indicated in the key provided next to the graph. Uncropped gels/blots for all the images in this figure are provided in the source data.

We additionally found that IN-binding-defective INI1 mutant (D225G) was unable to compete with TAR RNA to bind to IN (Fig. 5c, lane 4). However, mutant (D225E) with a conservative substitution, behaved like the wildtype and inhibited binding of TAR to IN (Fig. 5c, lane 5), suggesting that a negatively charged residue at 225 position is important for INI1 binding to IN. These results support the hypothesis that INI1 and TAR bind to IN and compete with each other in vivo.

To validate the IN-INI1 interaction model and to establish that INI1 interaction with IN is necessary for viral replication, we carried out complementation assay in INI1$^{-/-}$ MON cells. Previously we demonstrated that lack of INI1 leads to a defect in viral particle production and that co-expression of INI1 complements this defect[6,9]. We tested the function of INI1 mutants D225G and D225E on their ability to support particle production in INI1$^{-/-}$ MON cells. The results indicated that

there was a defect in particle production in the absence of INI1 (Fig. 5d, EV = Empty vector). Co-transfection of wild type INI1, but not the IN-interaction-defective INI1(D225G) mutant, lead to particle production (Fig. 5d). On the contrary, the INI1(D225E) mutant significantly increased the particle production compared to the empty vector (Fig. 5d). These results are consistent with the observation that D225G does not bind to IN, and support the hypothesis that INI1 binding to IN is required for particle production.

Finally, to determine the effect INI1-interaction defective mutations on particle morphogenesis, we produced W235E and R228A IN mutants of HIV-1$_{NL4-3}$ in 293 T cells and carried out transmission Electron microscopy (TEM) and Cryo-Electron tomography (Cryo-ET) studies (Fig. 6a and b). The EM structure

of R228A virus demonstrated the presence of empty capsids with unpackaged ribonucleoprotein (RNP) appearing as eccentric electron dense material in ~90% of virions (n = 186), consistent with it being a class II IN mutant[39] (Fig. 6a and Supplementary Fig. 8a). The Cryo-ET data indicated that W235E mutant particles (n = 130) also resembled other class II IN mutants and a WT sampling (n = 23) was consistent with previously characterized HIV-1 WT virion morphology[40,41] (Fig. 6b and Supplementary Fig. 8b). Compared to WT, W235E virions more frequently contained abnormal cores (54% versus 21%) or eccentric condensates of unpackaged RNP (68% versus 26%). W235E mutants were three-fold less likely to exhibit WT-like morphology, with RNP properly encapsidated in a conical core (23% versus 65%). Notably, conical capsids were over seven times

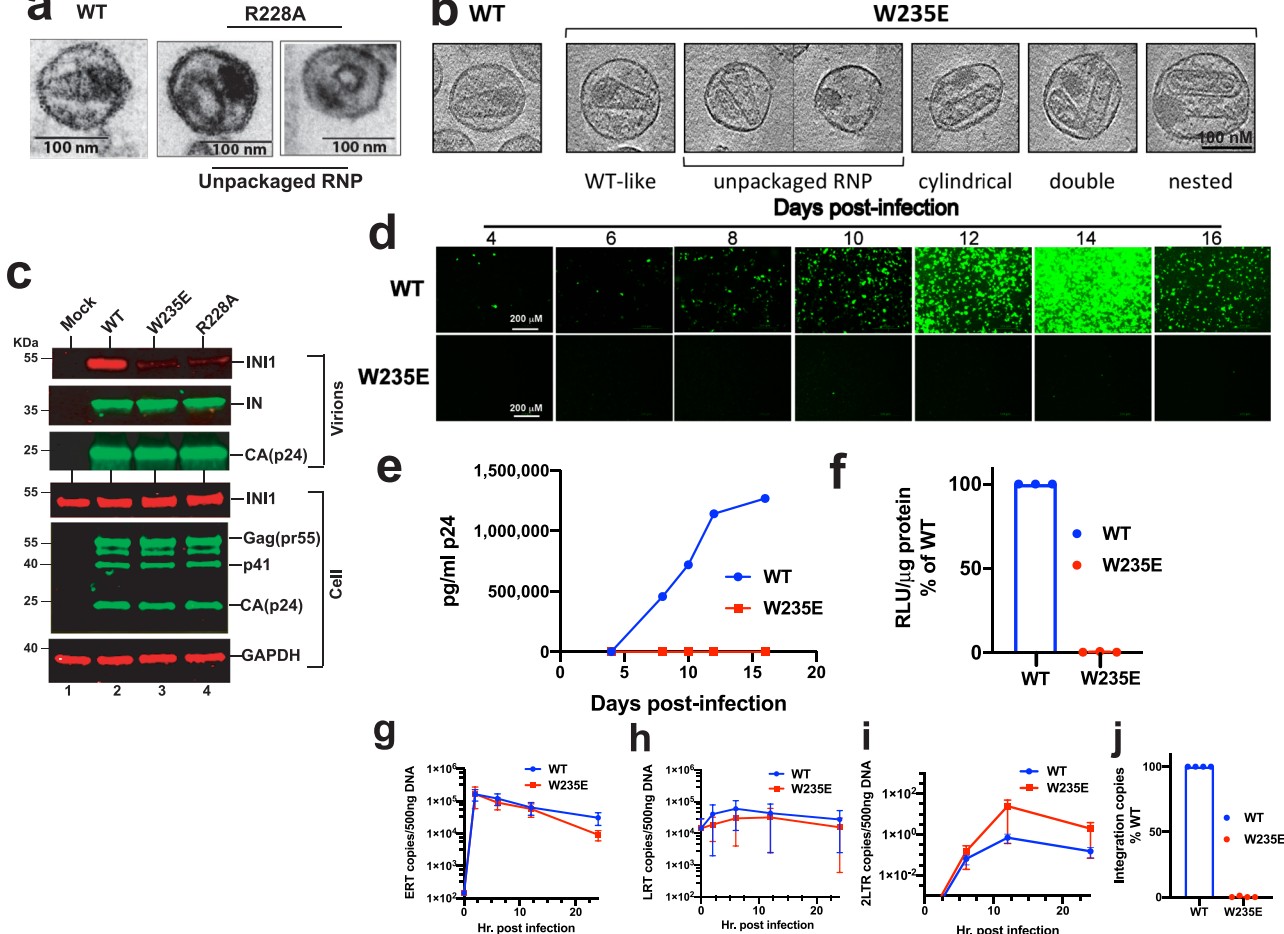

**Fig. 6 Particle morphology and replication of INI1-interaction defective IN mutants. a** TEM analysis of wild type (WT) and R228A mutant HIV-1NL4-3 particles. Note the empty capsids with unpackaged RNP in R228A mutant. **b** Cryo electron tomography (CryoET) studies to demonstrate the defect in particle morphology of the virions harboring W235E mutations. Leftmost panel indicates CryoET structure of wild type (WT) particle. The rest of the panels indicate various particle morphologies observed in mutant virions. **c** IN binding is necessary for INI1 incorporation into HIV-1 virions. The gel images are from one out of three independent experiments. Immunoblot analysis of concentrated WT, W235E, or R228A HIV-1$_{NL4-3}$ virions produced in 293 T cells. Top three panels represent immunoblot analysis of concentrated virions and the bottom three panels correspond to producer cell lysates. Western analysis was carried out using α-IN (to detect IN), α-p24 [to detect Capsid (CA, p24) and Gag (pr55)], α-BAF47 (to detect INI1) or α-GAPDH (as loading control) antibodies. Uncropped blots of this Western analysis are provided in the source data. **d–j** Analysis of replication of W235E mutant. **d** Fluorescence microscopy images of CEM-GFP cells infected with HIV-1$_{NL4-3}$ (25 ng p24 each) of WT or W235E mutant in a multiday infection. **e** Graphic illustration of virus particle release in the culture supernatant of the experiment in **d**, measured by p24 ELISA (Representative of two independent experiments). **f** Infectivity of HIV-1-Luc reporter virus harboring either a wild type (WT) or a W235E mutant integrase. The graph represents luciferase activity of infected cells, 24 hours post-infection (n = 3 independent experiments, Mean ± SEM). **g–i** Graphic representation of effect of W235E IN mutations on early RT products (**g**), late RT products (**h**), and two LTR circles (**i**), as measured by qRT-PCR, at indicated times, post-infection. The data represent average of three independent experiments (n = 3 independent experiments, Mean ± SEM). **j** Graphic representation of effect of W235E mutant on integration as measured by Alu-Gag PCR at 24 h post-infection. Data are compared to WT and represents average of three independent experiments (n = 3 independent experiments, Mean ± SEM). Uncropped gels and raw data are provided in the source data.

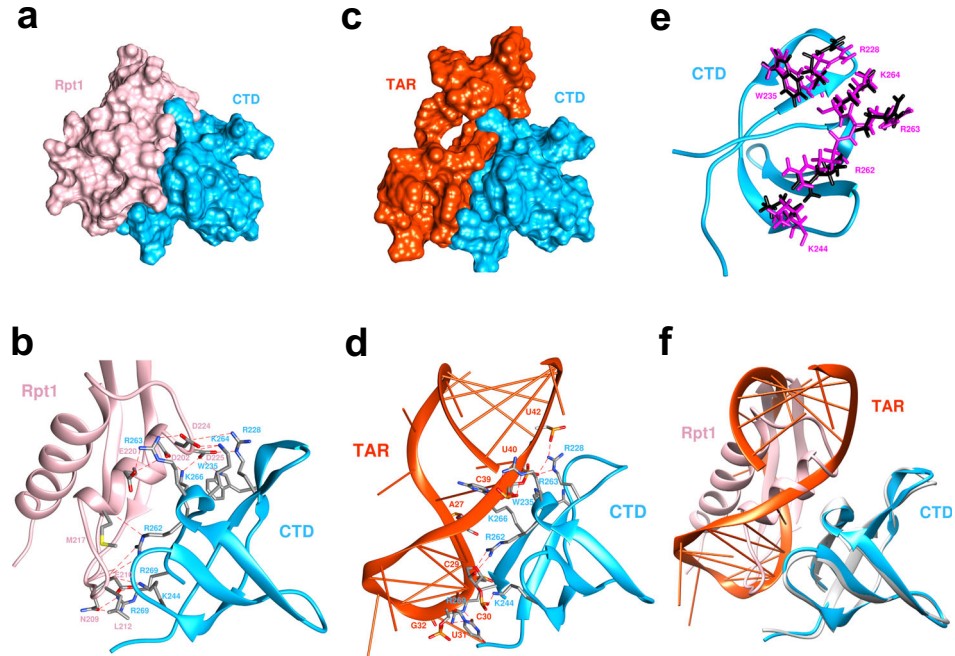

**Fig. 7 Molecular docking studies to compare Rpt1 INI1$_{183-304}$ and TAR binding to IN CTD. a** and **c** Surface representation of bound modeled complexes of CTD/Rpt1 and CTD/TAR. **b** and **d** Ribbon diagram showing interface residues of CTD/INI1-Rpt1 and CTD/TAR complexes. INI1-Rpt1 cartoon and residue labels are shown in pink, CTD cartoon and residue labels are shown in blue, and TAR cartoon and nucleotide labels are shown in reddish-orange. Note that interacting phosphate groups, bases, and sugars are shown. **e** Orientations of the key residues on CTD after docking shown in magenta (interacting with INI1-Rpt1) or black (interacting with TAR). **f** Superimposition of the CTD/Rpt1 and CTD/TAR complexes shows the identical orientation of CTD and nice overlap of Rpt1 and TAR RNA regions. In all panels, IN-CTD is represented in bright blue, INI1-Rpt1 in pink, TAR RNA in orange colors respectively. PDB files of models in this figure are included in Supplementary Data 1.

more likely to contain RNP, with 81 filled out of 112 (72%), compared to cylindrical capsids, with only 5 filled of 52 (9.6%) (Fig. 6b and Supplementary Fig. 8b). These findings together suggested that R228A and W235E mutations disrupted HIV IN role in particle morphogenesis.

It is known that INI1 is selectively incorporated into HIV-1 virions[10,11]. We tested to determine if INI1/IN interaction is necessary for INI1 incorporation into virions using W235E and R228A mutants. Concentrated virions and the corresponding producer cell lysates were subjected to Western analysis to detect viral proteins and INI1 (Fig. 6c). We found that wild type virus (Fig. 6c lane 2), but not the W235E and R228A mutant virions incorporated INI1, indicating that binding to IN is necessary for this incorporation (Fig. 6c lanes 3 and 4).

Further virological analysis of W235E mutants in both multiday and single cycle infection assays, investigating various post-entry events [Early RT(ERT), Late RT (LRT), nuclear localization and integration] confirmed several reports of a selective defect in integration in vivo[34,42] (Fig. 6d–j). Our virological studies provided additional information that the W235E mutant is defective for particle morphogenesis, while confirming previous results of a selective defect in integration.

**Modeling TAR/IN-CTD complex and comparative analysis reveals the structural basis of similarity of INI1$_{183-304}$ and TAR binding to IN-CTD.** While some of the IN-CTD residues important for HIV-1 RNA interaction are known, the IN-CTD/TAR complex structure is unknown[18]. To determine the structural basis of similarity in binding of INI1 and TAR, we generated a docking model for the IN-CTD/TAR interactions using IN-CTD [1QMC] and TAR [1ANR] structure, and further refined IN-CTD/INI1$_{183-304}$ complex model, this time by providing interaction restraints for the interface residues based on experimental data (see

Methods), using the in-house docking software, MDockPP[43] (Fig. 7a and b and Supplementary data 1). The docked complex with the lowest (best) score of ITScorePP[44,45] indicated that upon complex formation ∼865.0 Å$^2$ of the solvent-accessible surface was buried (Supplementary Table 4), which was larger than the buried surface area of the earlier IN-CTD/INI1$_{183-304}$ model and the comparable SWIRM/INI1 Rpt1 complex (699.1 Å$^2$,[35]). This second modeled complex structure was in agreement with the earlier model (Fig. 2), where the negatively charged residues from α-1 helix of Rpt1 directly interacted with positively charged residues of IN-CTD. We could identify a total of 14 hydrogen bonds formed across the interface (Fig. 7b and Fig. 2). The 6 residues on IN-CTD (R228, W235, K244, R262, R263, and K264) and 6 residues on INI1-Rpt1 (E210, L212, M217, E220, D224, and D225) constituted a strip of hydrogen-bond network, establishing binding specificity (Fig. 7b). In addition, the binding affinity was conferred by hydrophobic interactions between INI1-Rpt1 and IN-CTD (Supplementary Fig. 9a). On IN-CTD, a patch of the van der Waal surface that consists of a group of mostly hydrophobic residues (I220, F223, W235, A265, I267, and I268) matched the shape of a patch on Rpt1 that was also defined by mostly hydrophobic residues (L212, M213, F218, I221, and L222) (Supplementary Fig. 9a). The region of the hydrophobic interactions was encircled by the residues forming the hydrogen-bonding network.

To understand the similarity in interaction of INI1-Rpt1 and TAR RNA with IN-CTD, we generated a docking model of IN-CTD/TAR RNA complex by using the lowest energy conformer from the NMR structures of TAR [1ANR] and that of IN-CTD [1QMC] and by using MDockPP[43]. After applying the experimental restraints (see Methods), the top docked structure by reranking with ITScorePR[45] showed that the TAR RNA forms a complex with IN-CTD through the same interface region of IN-CTD as in the IN-CTD/INI1$_{183-304}$ complex, covering a similar

net surface area of 826 Å$^2$ (Fig. 7c, d and Supplementary Tables 4, 5 and Supplementary Data 1). This model indicated that the arginine and lysine residues of IN-CTD make multiple hydrogen bonds and electrostatic interactions with the phosphate backbone of TAR RNA (nucleotides 27–32 and 37–42) constituting the binding interface. IN-CTD specifically bound the minor groove of TAR RNA through 14 hydrogen bonds (similar to that of INI1-Rpt1) across the interface, including R228-U40, R228-U42, R262-C29, R263-C39, R269-U31, R269-G32, K244-C30, K266-A27, and K266-C39 (Fig. 7d). Meanwhile, the hydrophobic residues at the interface of IN-CTD protruded into the minor groove to form non-polar interaction with the hydrophobic part of the bases (Fig. 7d). The previous report indicated that nucleotide C30 of TAR and the loop were important for IN binding using CLIP-seq assay[18]. Furthermore, deletions, but not the nucleotide substitutions, of the loop and the adjacent three-nucleotide bulge of TAR reduced the IN binding, suggesting that IN prefers specific structural elements of TAR rather than specific nucleotide sequences[18]. Our model is consistent with this report and suggests that spatial positioning of the phosphate groups from the loop, the bulge, and the strand opposite of the loop in TAR RNA are important for interaction with IN residues.

When IN-CTD/TAR and IN-CTD/Rpt1 complexes were structurally aligned over IN-CTD, it became evident that both TAR and INI1-Rpt1 engage with the same binding interface and same residues of IN-CTD (Fig. 7e, f). There were only minor conformational differences in the side-chains in the IN-CTD interface residues, suggesting that no large side-chain rearrangements were required when switching from INI1-Rpt1 to TAR (Fig. 7e). The structural elements of INI1-Rpt1 and backbones of TAR seemed to perfectly line up in space (Fig. 7f). Moreover, the phosphate groups of TAR RNA were present in close proximity with the negatively charged residues of INI1-Rpt1 when the two docked complexes were superimposed (Supplementary Fig. 9b). In addition, similar buried solvent accessible surface areas, 865 Å$^2$ for IN-CTD/Rpt1 versus 826 Å$^2$ for IN-CTD/TAR, further supported that the two binding sites were the same for Rpt1 and TAR.

In the INI1-Rpt1 structure, the two β-sheets and helices of the INI1-Rpt1 (aa 183–248) come together to form a central hydrophobic barrel-like core, which is decorated by a string of negatively charged surface-exposed residues D192, E194, D196, E220, D224, D225 and D227 (Fig. 8a). Comparison of the arrangement of negatively charged residues of Rpt1 to the phosphate groups on the TAR RNA NMR structure (Fig. 8d), demonstrated a similar placement of negative charges on the two molecules. This observation, combined with the observation of the close proximity of the charged residues of INI1-Rpt1 with the phosphate residues of the TAR RNA when the two complexes were superimposed (Supplementary Fig. 9b), suggest that the INI1-Rpt1 domain may structurally mimic TAR RNA involved in binding to IN-CTD (Fig. 8b, c).

## Discussion

Here we report a finding that suggests structural mimicry between INI1 Rpt1 domain and TAR RNA. This finding rests on a series of structural and functional studies. First, docking of NMR structures revealed that the IN–CTD interface residues required for INI1$_{183-265}$ binding overlap with those required for TAR RNA association. Second, mutational analyses validated the predicted structure of the INI1$_{183-265}$/IN-CTD complex. Third, binding experiments confirmed the requirement of the same IN–CTD residues for interaction with INI1$_{183-265}$ and TAR, indicating that Rpt1 and TAR recognize the same IN-CTD surface. Fourth, INI1$_{183-265}$ and TAR competed with each other for IN–CTD binding in vitro and in vivo.

Finally, modeling the structure of IN–CTD/TAR complex suggested that the same positively charged residues of IN–CTD interacted with the negatively charged surface residues of INI1-Rpt1 and the phosphate groups of the TAR RNA. This was borne by the fact that when the NMR structures of the two molecules were compared, the distribution of negatively charged residues on INI1-Rpt1 surface were remarkably similar to the arrangement of the phosphate groups of interacting nucleotides on TAR. These studies together suggest that the Rpt1 domain of INI1 may be a TAR RNA mimic.

While Rpt1 (INI1$_{183-248}$) is a well-ordered structure, INI1 linker region (aa 249–265) is disordered. We propose that this region allows flexible positioning of Rpt1 relative to Rpt2 (aa 266–319), permitting Rpt1 to associate with various partners at different times, including IN and components of SWI/SNF. CryoEM studies of the intasome (the minimal unit for integration containing an IN tetramer with bound DNA) reveal that the IN-CTD domain exists in variable spatial positions in relation to CCD, due to the flexible linker region between CTD and CCD domains[46,47]. It is likely that INI1-Rpt1 can interact with some of the CTD domains within the intasome based on their spatial positioning. We propose that the interaction between INI1-Rpt1 and IN-CTD remains as predicted by our model within the full length IN and INI1 complex, permitted by the flexible linker regions in the two proteins. A future cryoET analysis of the full length IN/INI1 complex or the IN/INI1/TAR RNA will be needed to reveal the higher order structures of these molecules.

Our model reveals extensive side-chain interactions at the IN-CTD/INI1$_{183-304}$ interface. D224 and D225 residues, part of the D224–D225–D227 Asp triad in the INI1-Rpt1 α1 helix, form hydrogen-bonding interactions with basic residues R228, K264, and R263 of IN–CTD. The W235 residue of IN–CTD exhibits hydrophobic interactions with INI1$_{183-304}$. As predicted by our model, substituting W235 by charged residues W235E or W235K, but not by another aromatic side chain (W235F), disrupts IN/INI1 interaction and the integration activity of IN in vivo[35]. These studies not only validate the IN-CTD/INI1$_{183-304}$ complex model but suggests that IN/INI1 binding may be important for integration.

While RNA mimicry by INI1 Rpt1 is unexpected, nucleic acid mimicry by proteins exists. Prokaryotic elongation factor-P (EF-P) mimics tRNA$^{Asp}$ and facilitates the elongation of difficult-to-synthesize proteins by alleviating ribosomal stalling during translation[48] and RRF (ribosomal recycling factor) and EF-G proteins mimic tRNA to regulate various stages of translation[49]. Tumor suppressor p53 helix H2 mimics ssDNA and competes with it to bind to RPA (Replication protein A) 70 N complex to signal DNA damage[50]. Moreover, Shq1p, an assembly factor for the biogenesis of ribosomes in yeast, mimics RNA, binds to the RNA-binding domain of Cbf5p and operates as a Cbf5p chaperone and an RNA placeholder during RNP assembly[51].

What might be the role of TAR RNA mimicry by INI1-Rpt1 in HIV-1 replication? INI1 binds to IN within the context of GagPol and is incorporated into the virions in an IN-dependent manner[10,11] (and current data). Since INI1-Rpt1 and TAR bind to the same IN surface, we propose that these two interactions with IN could be separated by space and time. We also propose that INI1 binding provides a "place-holder" function for RNA binding during assembly. Binding of INI1 to GagPol during assembly may prevent premature binding of RNA to IN to prevent a possible steric hindrance (Fig. 8e, panel 1). Since INI1 and TAR RNA can compete with each other and bind to the same surface of IN, INI1 may compete off RNA binding to IN within GagPol. Class II-mutants defective for binding to RNA or INI1-interaction-defective-IN mutants within GagPol would be expected to fail to bind RNA, relieving the aforementioned steric

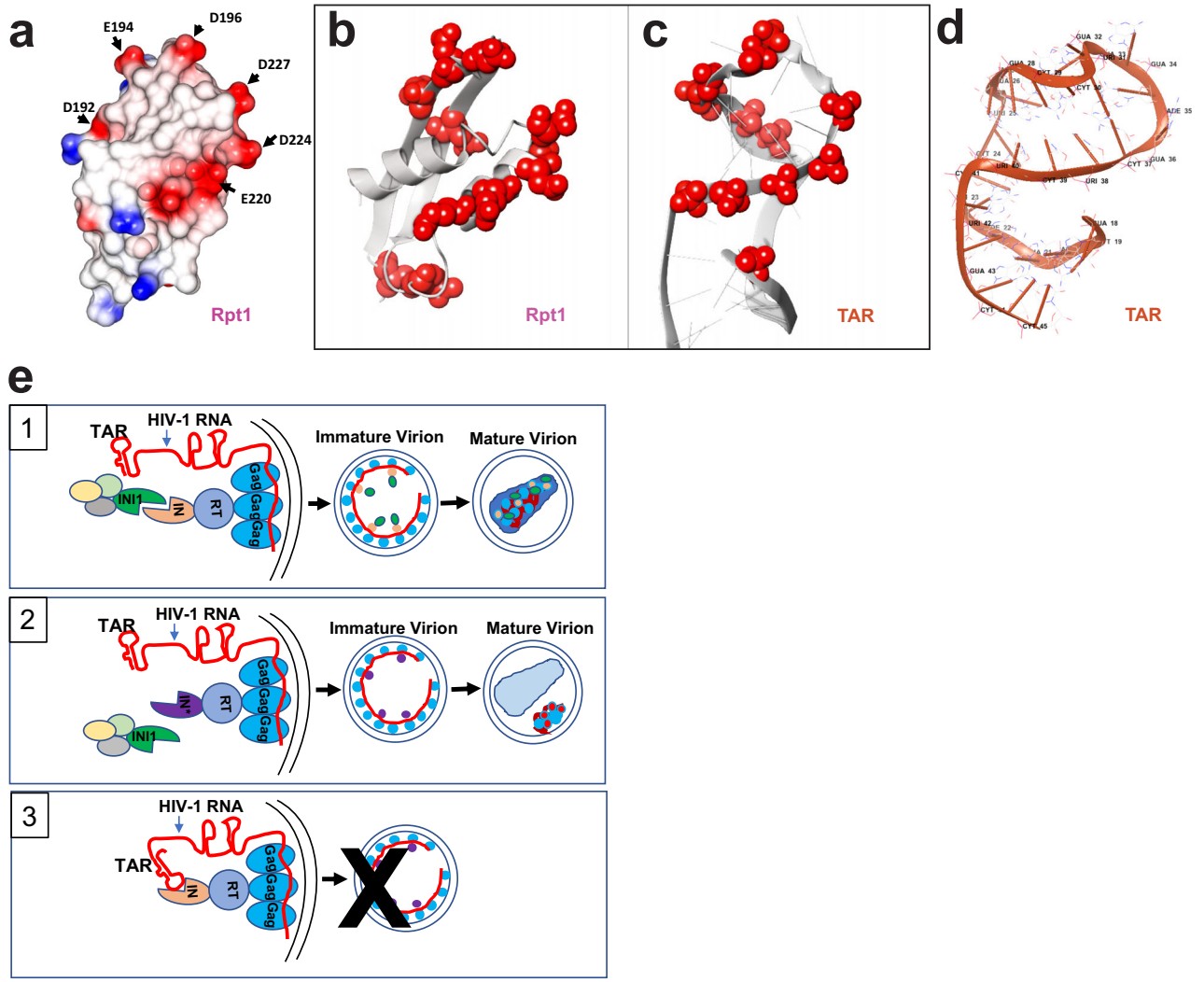

**Fig. 8 TAR RNA mimicry of Rpt1 domain and model for role of INI1 during particle production. a** Surface electrostatic computation of INI1$_{183-245}$ NMR structure indicating negatively (red) and positively (blue) charged and hydrophobic (white) residues. **b** and **c** Cartoon illustrating the similarity of INI1-Rpt1 to TAR RNA. Ribbon diagram of NMR structure of INI1 Rpt1 (left) where the side chains of all 11 negatively charged surface residues are depicted as red spheres, and that of TAR RNA where phosphate groups of the interacting nucleotides are depicted as red spheres. **d** Ribbon diagram of NMR structure of TAR RNA (PDB ID: 1ANR). **e** A model to understand the role of RNA mimicry of INI1 Rpt1 domain during HIV-1 assembly. Panel 1: In a producer cell where both INI1 and genomic RNA are present, INI1 acts as a place-holder and binds to IN portion of GagPol to prevent RNA binding to it, which otherwise may cause steric hindrance. Both RNA and INI1 are incorporated into the virions resulting in correct particle morphogenesis. Panel 2: RNA-interaction-defective and INI1-interaction-defective mutants of IN are impaired for binding to both RNA and INI1 and hence there is no steric hindrance for assembling GagPol. However, during particle maturation, lack of binding to RNA and/or INI1 could lead to morphologically defective particles, as shown in the empty conical capsid and unpackaged materials on the side of the capsid in the virion. Panel 3: Lack of INI1 leads to binding of RNA to IN portion of GagPol, which results in defective assembly and particle production. Gag (bright blue), RT (Reverse transcriptase; in blue), IN (wheat color), IN mutant (purple), INI1 (green), and HIV-1 RNA with TAR (red) are represented with the same colors both in the cells and in the virion particles, as indicated. Yellow, light green, and gray ovals represent possible INI1-binding proteins.

hindrance permitting assembly (Fig. 8e, panel 2). Thus, INI1 may act as a "place-holder", which would be critical for assembly and particle production. Indeed, lack of INI1 inhibits assembly and particle production[6,9,11] (Fig. 8e, panel 3). Once assembly is completed and upon Gag proteolysis, INI1 binding may be displaced by RNA binding to IN during particle maturation. Since INI1 and RNA binding surfaces on IN overlap, it is not possible to distinguish between the roles of these two interacting partners during particle maturation within the virions at this point.

It is conceivable that the suggested mimicry between TAR and INI1-Rpt1 domain may have first evolved to modulate HIV-1 transcription, and the effects on assembly are rather consequential. INI1 directly binds to acetylated Tat[19], and together

with the SWI/SNF complex, it facilitates Tat-mediated transcriptional elongation[12,19,52]. It is possible that the suggested TAR mimicry of INI1-Rpt1 domain allows it bind to Tat, which may be important for recruiting SWI/SNF complex to sites of LTR transcription to facilitate elongation[52]. Future experiments on TAR RNA mimicry of INI1 is likely to lead to a better understanding of Tat/INI1 and IN/INI1 interactions and may lead to the development of a unique class of drugs to simultaneously inhibit multiple stages of HIV-1 replication.

Since INI1–Rpt1 domain is highly conserved among eukaryotes and HIV-1 is of recent origin, it is likely that Rpt1 domain has evolved to mimic a cellular RNA, rather than HIV-1 RNA. TAR RNA mimics cellular 7SK RNA at the SL1 (stem loop 1)[53].

Therefore, it is plausible that INI1-Rpt1 may have evolved to mimic 7SK. Thus, future experiments to understand the possible INI1-Rpt1 RNA mimicry are likely to unravel insights about INI1 role not only in HIV-1 replication, but also in cellular transcription and tumor suppressor function.

## Methods

**Cloning of INI1₁₈₃₋₂₆₅, INI1₁₈₃₋₃₀₄, and IN fragments**. Fragments containing INI1$_{183-265}$ and INI1$_{183-304}$ were cloned into pET28a-h6-smt3 vector using sequence and ligation-independent cloning (SLIC) method[54]. Briefly, vector sequences were PCR-amplified using the primers, VFOR and VREV (Supplementary Table 6a). INI1$_{183-265}$ and INI1$_{183-304}$ fragments were amplified using the forward primer S6(Rpt1)-For and two different reverse primers, INI1(aa 265)-Rev and INI1(aa 304)-Rev, respectively (Supplementary Table 6a). PCR was performed using Phusion Polymerase followed by digestion of PCR amplified fragment with DpnI for 2–4 h or overnight at 37 °C, purified using Qiagen PCR purification kit (Catalogue # 28104) and then gel purified (Catalogue # 28704). The gel-purified fragments were subjected to T4 DNA polymerase reaction to generate single stranded overhangs in the absence of dNTPs at room temperature for 30 m followed by quenching the reaction by the addition of 0.5 mM dNTP and immediately heating at 65 °C for 10 m to deactivate the T4 enzyme. The SLIC reaction is set up by mixing 1:3 molar ratio of vector to insert, 10x ligation buffer, and 20 mM ATP and incubated at 37 °C for 30 m. A total of 1–2 μl of the reaction mixture was then transformed into *E.coli*. The resulting transformants were sequenced to confirm the presence of INI1 insert using T4 terminator primer (Novagen Catalogue # 69337 5 p mole/μl).

To clone GST-IN, GST-NTD, GST-CCD, and GST-CTD, fragments encoding the three IN domains were PCR amplified using the HIV-1$_{Hx3B}$ DNA as a template, and were inserted into pGEX3xPL vector to obtain the respective GST-fusion expression constructs.

**Expression and purification of INI1₁₈₃₋₂₆₅**. Expression of His$_6$-SUMO-INI1$_{183-265}$ protein from the plasmid (pET28a-h6-smt3-INI1$_{183-265}$) in *E.coli* was confirmed by immunoblot analysis using α-6His [Clontech, Catalogue # 631212; Lot# 8071803; 1:1000 dilution) and α-BAF47 (BD Transduction laboratories, Catalogue # 612110; Lot # 7144795; 1:1000 dilution) antibodies. *E. coli* strain BL21(DE3)lysS harboring the expression plasmid was induced with 1 mM IPTG in lysis buffer (25 mM HEPES, pH 7.4, 10% glycerol, 1 mM PMSF and 0.1% Triton X-100), and subjected to sonication. The sonicated culture was rocked for 45 m, clarified by centrifugation and was loaded on to pre-equilibrated Ni-NTA column in buffer (1 mM HEPES, pH 7.4, 10% glycerol, 0.5 M NaCl and 30 mM Imidazole). The bound proteins were washed with several volumes of the same buffer and eluted in 25 mM HEPES, pH 7.5, 10% glycerol, 0.5 M NaCl, and 300 mM imidazole. The eluted protein was digested with SUMO protease for ~16 + h at 4 °C with rocking. After proteolysis, the buffer was exchanged with holding buffer (25 mM HEPES, pH 7.4, 10% glycerol, and 0.25 M NaCl), by spinning with Amicon Ultra-15 Centrifugal Filter unit. The His$_6$-SUMO tag was removed by running protein on Ni-NTA column equilibrated with holding buffer. Finally, the eluted and purified protein was passed through 16/60 Superdex 200 gel filtration column, using the buffer, 25 mM HEPES, pH 7.4, 0.25 M NaCl, 2 mM DTT, and 5% glycerol. The protein eluted as a single peak and was collected and dialyzed in the buffer 25 mM HEPES, pH 7.4, 5% glycerol, 0.15 M NaCl, 1 mM EDTA, and 2 mM DTT.

To purify the labeled INI1$_{183-265}$ for NMR studies, Rosetta (DE3) cells (Novagen) were transformed with pET28a-h6-smt3-INI1$_{183-265}$ (clone 3.1) expression plasmid and the cultures were grown at 37°C in 1 L of minimal medium supplemented with 1 g $^{15}$NH$_4$Cl and 2 g $^{13}$C-glucose (Cambridge Isotope Laboratories). The bacteria were induced at a cell density of OD$_{600}$ 0.6 by the addition of 0.5 mM IPTG and were then incubated at 22 °C overnight. The cells were pelleted by centrifugation at 7,000 g for 15 minutes and the pellets were stored at -80 °C for further processing. The pellets were thawed and resuspended in lysis buffer [20 mM HEPES pH 7.6, 500 mM NaCl, 20 mM imidazole, 10% glycerol containing 0.2 mM 4-(2-aminoethyl) benzene sulfonyl fluoride hydrochloride (AEBSF) and 4 units/ml of DNase I]. The cells were lysed using an Emulsiflex C3 (Avestin) and the lysate was centrifuged at 17,000 g for 1 hr at 4 °C. The supernatant was applied to a prepacked His60 superflow column (Clontech) using an ÄKTA avant purification system (GE Healthcare). The column washed with 10 column volumes (CV) of lysis buffer without protease inhibitor and DNase. The protein was eluted with 20 mM HEPES pH 7.6, 500 mM NaCl, 500 mM imidazole, 10% glycerol, and then loaded onto a Superdex 200 16/60 column (GE Healthcare) in SEC buffer (20 mM HEPES pH 7.6, 150 mM NaCl, 5% glycerol and 10 mM DTT). The fractions containing the protein of interest were pooled together and digested with SUMO hydrolase (ratio 100 to 1 respectively) overnight at 4 °C. The his-smt3 tag was removed by loading the digested proteins onto a prepacked Ni Sepharose high performance column equilibrated in 20 mM HEPES pH 7.6, 150 mM NaCl, 5% glycerol, and 10 mM DTT. The column was washed with 3CV of SEC buffer. The flow-through and the washes containing the protein of interest were pooled together and concentrated using a Vivaspin 20 filter with a 3 kDa cut-off (Sartorius AG). The protein was finally loaded into a Superdex75 16/90 column (GE Healthcare) in NMR buffer (10 mM sodium phosphate pH 6.8, 150 mM NaCl,

1 mM EDTA, 5 mM TCEP). The fractions containing the protein were concentrated using a Vivaspin 20 filter with a 3 kDa cut-off (Sartorius AG) and snap frozen for storage at -80 °C. SDS-PAGE was used to determine the sample purity and the correct identity of the purified protein was achieved by LC-MS/MS.

**Nuclear magnetic resonance**. Isotopically enriched INI1$_{183-265}$ was purified and concentrated to 1 mM in 150 mM NaCl, 10 mM sodium phosphate, 10 % D$_2$O, and 5 mM BME (pH 7.4). All NMR data were acquired at 25°C on a 600 MHz cryoprobe-equipped Agilent instrument, or at 900 MHz on a Bruker Avance II. NMR data was collected using manufacturer's TopSpin v2.3 operating software.

Sequence specific and side-chain assignments were obtained by standard nD triple resonance methods. All 3D experiments were acquired as non-uniform sampled experiments using the MDDNMR v2.1 approach[55]. All NMR data sets were processed with nmrPipe/nmrDraw[56] and analyzed using CCPN Analysis 2.3[57]. Chemical shifts were indirectly referenced to 4,4-dimethyl-4-silapentane-1-sulfonic acid (DSS). Interproton distance restraints were derived from 3D $^{15}$N- and $^{13}$C-edited NOESY-HSQC spectra with a mixing time of 150 ms. The full table of characterization is reported in supplementary material.

**Analytical ultracentrifugation**. Sedimentation velocity analyses were conducted in a Beckman XL-I analytical ultracentrifuge at 270,789 × g and 25 °C using double sector centerpieces in the An60 rotor. Sedimentation was tracked using the absorption optics at 280 nm. The buffer was 20 mM HEPES pH 7.5, 300 mM NaCl, 2 mM DTT, and 5 % glycerol. The reported sedimentation parameters were corrected to standard conditions (20, w) using values of the partial specific volume = 0.742, solvent density = 1.02 g/mL, and solvent viscosity = 1.16 cp. The data were analyzed using the time-derivative method implemented in the program DCDT + [58,59]. The best fit values and 68.3% confidence limits are reported.

**Protein structure modeling and analysis**. Robetta was used to builds various models for Rpt1-Rpt2 and 183-304 fragments of INI1. Robetta generates three- and nine-residue fragment libraries that represent local conformations seen in the PDB, and then assembles models by fragment insertion to form low-energy global structures. Detailed methodology of Robetta can be accessed from[25] and http://robetta.bakerlab.org/. To assess the overall stereochemical quality of the generated 3D model, the geometrical accuracy of the residues and 3D profile quality index were inspected with the PROCHECK (ver. 3.5)[26]. Additionally, WHATIF (ver. 8.0) modeling package software was used to analyze the quality of model by checking clashes[60]. The modeled protein is also validated by VERIFY3D, which checks compatibility of 3D models with its sequences[28]. The statistics of non-bonded interactions between different atom types were detected and value of the error function was analyzed by ERRAT[61]. PROSA was used for final model to check energy criteria[29].

**Molecular docking studies**. First, the INI1$_{183-304}$ fragment and CTD were docked using HADDOCK v2.4[62]. The docking was done ab initio, without providing interaction restraints; instead center-of-mass restraints were provided to guide the docking. The center-of-mass restraints are automatically generated by calculating the dimensions of each molecule along the x, y and z axes ($d_x, d_y, d_z$) and summing the average of the two smallest components per molecule. The resulting distance was used to define a restraint between the center of mass of each subunit with an additional upper bound correction of 1 Å. Finally, clustering was done to sort the docking solutions. The center of the best cluster was taken as the final model and was analyzed further in detail.

Second, both INI1$_{183-304}$ fragment and TAR RNA were docked separately onto IN-CTD using MDockPP[43]. The docking process was performed by heavily sampling the relative binding orientations, creating 54,000 putative binding poses. These binding poses were screened by the following experimental constraints: The residues W235, R228, K264, K266, and R269 of CTD were required to be within 5 Å of INI1$_{183-304}$ fragment and TAR RNA, respectively. For the INI1-Rpt1/IN-CTD structure prediction, an extra constraint was imposed that D225 of INI1-Rpt1 was within 5 A of IN-CTD. The surviving poses were then rescored with ITScorePP[44,45], and the top-ranking pose was selected and optimized using UCSF Chimera[63] (http://www.cgl.ucsf.edu/chimera). Structure visualization, structure characterization and analysis, and image rendering were carried out using CCP4mg (https://www.ccp4.ac.uk/MG/), MacPymol (PyMOL v1.7.6.4 Enhanced for Mac OS X, https://pymol.org/), Maestro (Schrodinger, https://www.schrodinger.com/maestro), and UCSF Chimera (version 1.14 at https://www.cgl.ucsf.edu/chimera).

**Generation of substitution mutations**. Mutagenesis was carried out using QuickChange Lightning site-directed mutagenesis kit (Agilent Catalog # 210518).

For introducing mutations into INI1$_{183-304}$ fragment, pET28a-h6-smt3-INI1$_{183-304}$ plasmid was used as template. Primers used for mutagenesis are provided in the Supplementary Table 6b. For introducing mutations into the CTD domain of IN, the plasmids pGEX3x-IN and pGST-IN-CTD(aa 201–288) were used as templates and the primers used for mutagenesis are provided in the Supplementary Table 6c.

The mutant clones were sequenced using the T7 terminator primers for pET clones and GST-forward and reverse primer for GST clones to confirm the presence of mutations in the sequence.

**Generation of W235E, R228A, and K244A mutant viruses**. HIV-1$_{NL4-3}$ viral clones containing IN mutations were generated in two stages. First, a 2.3 Kb Age I to Sal I (position 3485 bp to 5785 bp) fragment of HIV-1$_{NL4-3}$ containing the IN open reading frame was subcloned into the pEGFPN1 vector to generate an intermediate vector pEGFPN1-IN-int. IN mutations were introduced into pEGFPN1-IN-int using the QuikChange II Mutagenesis Kit (Stratagene). The Sal I/ Age I fragment containing the desired mutation was subsequently cloned into pNL4-3 and in some cases into pNL4-3.Luc.R-E-[HIV-Luc] to obtain mutant viral clones for use in multiday and single cycle infection assays.

**GST pull-down assays**. 5μg of GST-IN or GST-CTD and their mutant proteins bound to glutathione sepharose 4B beads were incubated for 1 hour at 4°C with normalized amounts of clarified bacterial cells lysates containing His$_6$-SUMO-INI1$_{183-304}$ WT and its mutants in binding buffer (20 mM HEPES- pH 8.0, 5 mM DTT, 0.5% IGEPAL, 200 mM NaCl, and protease inhibitor tablet Roche Catalogue # 11836170001). Following incubation, beads were washed 5-7 times with buffer containing 50 mM Tris-Cl pH 8.0, 1 mM EDTA, 500 mM NaCl, 0.5% IGEPAL, 25 mM PMSF. Bound proteins were separated by SDS-PAGE and analyzed using α-BAF47 antibody to detect bound INI1 or 6His-SUMO-INI1$_{183-304}$ proteins.

**AlphaScreen proximity assay to detect the protein-protein and protein-RNA interactions**. AlphaScreen assay was carried out using PerkinElmer Alpha-Lisa-6His Acceptor beads (Catalogue # AL128C), Alpha-Screen-GST donor beads (Catalogue # 6765300), GST-tagged and His-tagged proteins for protein-protein interactions; and Alpha-Screen Streptavidin donor beads (Catalogue # 6760002 S), Alpha-Lisa-GST acceptor beads (Catalogue # AL110C), Biotin-labeled RNA and GST-tagged proteins for protein-RNA interactions. The reaction was carried out in a reaction buffer contacting (25 mM HEPES pH 7.4, 100 mM NaCl (or 500 mM NaCl), 1 mM DTT, 1 mM MgCl$_2$ and BSA 1 mg ml/ml). The mixture of proteins or protein and RNA were incubated for 1 hr at room temperate with shaking in a multi micro-plate shaker and then 20 μg ml/ml accepter beads were added and incubated for additional 1 h with shaking. Finally, 20 μg ml$^{-1}$ of donor beads were added and further incubated for 1 hr. The Alpha score readings were measured using Alpha-compatible Envision 2105 multi-plate reader (Perkin-Elmer). All the experiments related to this method were carried out 3 to 5 times and data were analyzed by using Graph Pad Prism 9 version 9.0.0 (GraphPad Software, CA). The RNA sequences used in this assay are listed in the Supplementary Table 6d and were obtained from Integrated DNA technology.

**Co-IP and RNA-co-IP to determine interaction of IN, INI1, and TAR RNA in vivo**. For co-IP studies, pYFP-IN or mutants and pCGN-INI1 (expressing HA-INI1) were transfected into MON(INI1$^{-/-}$) cells. The transfected cells were lysed in 20 mM Tris-HCL (pH 8), 150 mM NaCl, 1% Triton X 100, 2 mM EDTA, 40 μL/ml Protease cocktail inhibitor (Roche cat no: 11836170001) and 40 μL/ml RNASe inhibitor (Invitrogen Cat. No: 10777019). The lysates were pre-immunoprecipitated with isotype IgG antibodies (Santa Cruise Biotechnology, Catalogue # SC-51993; Lot # F0316, 5μg/sample) and then immunoprecipitated using α-HA (Santa Cruz, Catalogue # SC-7392; Lot # L1218; 5μg/sample) antibodies overnight with shaking at 4 C. The immunocomplex was subjected to Western blot analysis using either α-IN (NIH AIDS reagent and repository, Catalogue # 3514; Lot # 130353, 1:500 dilution) or α-BAF47 antibodies.

For RNA-co-IP, MON cells were transfected with pCMV-Tat and pLTR-luc plasmids along with pYFP-IN and pCGN-INI1. Immunopreciptation was carried out as above using α-GFP (Cell Signal, Catalogue # 2555; Lot # 8; 5μg/sample) antibodies. The immunocomplexes were then split into two and RNA was isolated from one half using Trizol reagent (Invitrogen Catalogue # 15596026) and subjected to qRT-PCR using Early RT primers. The other half was subjected to immunoblot analysis.

As a negative control for IP and RNA-co-IP, isotype specific IgG (Santa Cruise Biotechnology, Catalogue # SC-51993; Lot # F0316, 5μg/sample) antibodies were used.

**Virus production and multiday infection**. HIV-1$_{NL4-3}$ viral stocks of wild type, W235E and R228A mutants were prepared by transient transfection of 30–40% confluent 293 T cells in 10 cm$^2$ plates with 10 μg viral DNA. Virus was collected 48 h post-transfection and clarified by passing through a 0.45 μm cellulose acetate filter (Corning). Clarified supernatant was treated for 30 m with 20 units/mL of DNase I (Roche Catalogue # 04716788001) at 37 °C. For the purpose of CryoET, the virions were concentrated using sucrose step gradient[64]. Viral stocks were measured for p24 using a p24 enzyme-linked immunosorbent assay (ELISA) (Advanced Bioscience Laboratories).

For the purpose of multiday infections, 25 ng p24 of each HIV-1$_{NL4-3}$ wild type or W235E mutant viruses were incubated with 200,000 CEM-GFP cells for 2 h in a 2 mL culture. After incubation with virus, 18 mL of complete RPMI1640 was added to the culture in a 75 cm$^2$ flask and incubated at 37 °C for several days. 1 mL of

culture was collected every two days for 16 days. Viral replication was monitored by observing cellular GFP levels and by measuring p24 levels in the culture supernatant.

**Real-time PCR to detect early and late RT products, 2LTR circles, and integrated product**. A total of 50% confluent 293 T cells were spinoculated with 20 ng p24 of HIV-Luc virus for 2 h at 15 °C and spun at 400 g. Cells were collected at various time points post-infection and harvested for genomic DNA using the DNeasy Blood and Tissue Kit (QIAGEN). About 300–500 ng genomic DNA was used per 50 μL real-time PCR reaction containing 300 nM primers, 100 nM probe and 2X Taqman Universal Master Mix (Applied Biosystems). The primers and probes that were used to detect early and late RT products[65] are listed in the Supplementary Table 6e. Cycling conditions on an ABI 7900HT at 2 m at 50 °C, 10 m at 95 °C, followed by 40 cycles of 15 s at 95 °C and 1 m at 60 °C.

To detect the integrated provirus, nested Alu-PCR was performed[66]. The genomic DNA was isolated 24 hours post-infection. In the first semi-quantitative PCR round, Alu-gag was amplified using the primers listed in Supplementary Table 6e. The Alu-gag PCR product was then subjected to a second round of qPCR using the late RT primers (MH531, MH532), and LRT probe (Supplementary Table 6e) using conditions described above. Standard curves were generated by isolating genomic DNA from 293 T cells stably infected with HIV-1 containing GFP and a hygromycin-resistance marker[66] and by performing nested PCR from this cell line. Ct values were then plotted against dilutions of the hygromycin-resistant HIV-1 DNA to create a standard curve.

**Electron microscopy**. Monolayer 293 T cell cultures producing HIV-1$_{NL4-3}$ WT or R228A mutant were fixed with 2.5% glutaraldehyde, in 0.1 M sodium cacodylate buffer, post-fixed with 1% osmium tetroxide followed by 2% uranyl acetate, dehydrated through a graded series of ethanol, cells lifted from the monolayer with propylene oxide and embedded as a loose pellet in LX112 resin (LADD Research Industries, Burlington VT) in eppendorf tubes. Ultrathin sections were cut on a LeicaUltracut UC7 (Leica Microsystems Inc., Buffalo Grove, IL), stained with uranyl acetate followed by lead citrate and viewed on a JEOL JEM 1400Plus transmission electron microscope (JEOL USA, Peabody, MA) at 120 kV.

**Cryo-ET**. Grids of HIV-1 WT and W235E virions were prepared as follows[67]. Purified fixed virus was mixed (2:1) with a suspension of colloidal gold particles (Electron Microscopy Sciences), applied to glow-discharged Quantifoil R2/2 200-mesh holey carbon grids (Structure Probe, Inc.), blotted, and plunge-frozen using a Leica EM GP (Leica Microsystems). For data acquisition, grids were transferred to a cryo-holder (type 914; Gatan), and single-axis tilt series were recorded at 200 keV on a JEOL 2200FS electron microscope equipped with an in-column energy filter (20 eV energy slit). Images were acquired on a K2 Summit direct electron detector (Gatan) in super-resolution mode at nominal 10 K magnification, giving a super-resolution sampling rate of 1.83 Å/pixel. Using SerialEM[68], dose-fractionated projections were typically acquired at 2° intervals from −56° to + 56° with a target defocus of -2.5 μm. The electron dose per exposure (five 0.2 s frames each) was ~1.3 e$^-$/Å$^2$, giving a total cumulative dose of ~75 e$^-$/Å$^2$.

For Image Processing and Analysis, Unbinned frames were aligned and averaged using the bseries function in Bsoft[69], and the motion-corrected tilt-series images were binned by four. Tomograms were reconstructed using Bsoft, and virions were extracted and denoised by 20 iterations of anisotropic nonlinear diffusion[70]. Denoised particles were then manually inspected for defects in assembly.

**Statistics and reproducibility**. Data are presented as mean value ± SEM (standard error of mean), calculated using GraphPad Prism 9 version 9.0.0. All experiments were conducted a minimum of three independent times using three independent preparations of the same protein or virus and were found to be reproducible. N values are indicated within figure legends and refer to biological replicates (independent experiments conducted using different preparations of proteins or viruses).

**Reporting summary**. Further information on research design is available in the Nature Research Reporting Summary linked to this article.

## Data availability

A reporting summary for this article is available as Supplementary Information file. The main data supporting the findings of this study are available within the article and its Supplementary Figures. The structures in the Fig.1b, c and Fig.8a, b are derived from the NMR structure deposited in the PDB database with ID:6AX5 (https://www.rcsb.org/structure/6AX5). The structural data for PDB IDs: 5I7A, 5I7B, 5GJK, 6LTJ, 1QMC, and 1ANR used in Fig. 1d are available in the PDB database (https://www.rcsb.org/). The PDB files of the modeling and docking data underlying Figs. 1e, 2, and 7 are provided as Supplementary data 1. The source data underlying Figs. 3–5 and 6c–j are provided as a Source Data file. The data in Fig. 6b are deposited in the EMBD database (under entry IDs EMD-22410 and EMD-22411). MDockPP structure prediction software used for the prediction of structures of docked complexes in the Fig. 7 is

accessible at http://zougrouptoolkit.missouri.edu/MDockPP. Supplementary data 1 and Source data are provided with this paper.

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

## Acknowledgements

This study was supported by NIH grants GM112520-02 and GM112520-04S1 to GVK; R01GM109980 to XZ; and NIAMS intramural research program to A.S. It was supported in part by the Einstein-Rockefeller-CUNY Center for AIDS Research (P30-AI124414). SM was supported by an Institutional training grant, T32-AI007501. NMR studies were supported by NCI core Center Grant P30 CA013330 and 1S10OD016305-01A1 the 900-MHz system was purchased with funds from NIH GM66354, the Keck Foundation and the New York Structural Biology Center. TEM studies were funded by NCI core Center Grant P30 CA013330 and the Shared Instrumentation Grant (SIG #1S10OD016214-01A1).

We thank Drs. Prasad, Kielian, Chandran, Wilson, and Query for critically reading the manuscript; Dr. Xiao Heng for the expert analysis and useful discussion on the structures; Timothy Mendez, Leslie Cummins, and Frank Macaluso for assistance with TEM; and Drs. Dennis Winkler and Bernard Heymann for cryo-ET resource, image processing, and expert advice. We thank the following for the generous gift of reagents - Drs. Ott (α-p24 antibody), Muesing and Mohammed (2LTR circle standard plasmid), Dreyfuss (GST-GEMIN2), Engelman (pGST-LEDGF), and Craigie and Lee (6His-Sso7d-IN construct). GVK is a recipient of Irma T. Hirchl/Monique Weill-Caulier scholar award and currently a Mark Trauner Faculty scholar in Neuro-Oncology.

## Author contributions

Conceptualization, G.V.K.; Methodology, S.B., L.Q. and X. Z. (Modeling), M.G., S.C., R. H., D.C. (N.M.R.), A.S. (Cryo-ET), G.V.K., and U.D. (Virological, biochemical and all other studies); Investigation, U.D., S.B., X.W., L.Q., M.S., S.M., R.H., L.A., S.C., R.P., P.R. K., M.N., S.A.A., M.B., S.A., X.Z., A.S., and D.C.; Writing-Original Draft, G.V.K. with contribution from the other authors; Writing-Reviewing and Editing, G.V.K.; Supervision, G.V.K.; Funding Acquisition, G.V.K.

## Competing interests

The authors declare no competing interests.
