## [Peer Review File · Nature Communications]

Reviewers' Comments:

Reviewer #1:

Remarks to the Author:

Bhutoria et al. present a very interesting and likely important study on how INI1 (syn hSNF5, BAF47, SMARCB1) interacts with HIV-1 IN to facilitate viral replication. Their findings suggest that the Rpt1 domain of INI1 mimics the TAR binding to HIV-1 IN CTD. The paper is well written and presents many results that support the authors' conclusion.

However, some parts of the paper may benefit from additional investigations/analyses that I feel would clarify and enhance their findings.

1. In Fig 1, the authors provide NMR structure of INI1 Rpt1 domain. Here, the authors should superimpose their structure and the crystal structure of INI1 Rpt1 domain and BAF155 SWIRM heterodimer [PDB ID: 5GJK; Yan et al JMB 2017]. Also, please check the NMR structure of Rpt1 [PDB ID: 5L7A/B; Sammak et al., FEBS 2018] and discuss the similarities and the differences.

2. In Fig 2, the authors provide a model of the IN CTD:INI1 Rpt1 interaction. HIV-1 IN exists in dimer↔tetramer equilibrium in solution. Can the authors discuss their model in this context, including the limitations of their model? Importantly, the authors should carefully evaluate the crystal structure reported by Yan et al., JMB 2017 and comment on the interaction surfaces overlap for HIV-1 IN and BAF155 SWIRM on INI1. Do IN and BAF155 SWIRM compete for INI1?

3. In Suppl. Fig S4F, the authors showed in vitro pull-down data. The full-length IN binds significantly better than the two domains tested. This begs for additional quantitative data and the authors should also show:

a. The interactions in cells by co-IP. Co-transfect INI1 and IN (WT or the different mutants tested in Fig 3) in the full-length context and show the binding efficiency.

b. The authors mentioned in the introduction that the stoichiometry of IN:INI1 in virions is ~1:2; however, it is not known whether that stoichiometry is a true reflection of the binary interaction in vivo. The authors should provide quantitative data using techniques such as analytical ultracentrifugation analysis to determine the stoichiometry of IN:INI1 complex.

4. Results shown in Figs 3C, 4, and 5 primarily focus on the IN determinants of the IN:INI1 interaction. Others have shown similar phenotype using either ALLINIs/LEDGINS/NCINs or class II IN mutants. The class II IN mutants have effect on IN multimerization, the IN:TAR interaction, and hence perturb virion morphogenesis resulting in aberrant morphology (Gupta et al PLoS Biol 2016; Kessl et al., Cell 2016). Can the authors rule out whether the effect of the IN mutants described here is not due to IN multimerization defect?

5. The experimental results shown in Fig 5 are tricky to understand. Based on the data shown in Fig 5A and several previous reports, these defective viruses are expected to be completely dead for completion of reverse transcription; however, the authors showed that the defect is only at the integration step (Fig 5E-H). Can they comment on this discrepancy?

Minor:

- Page 4: Replace the PDB ID: 5GJK with your own structure PDB ID: 6AX5.

- Ref #29 was cited for the "ββα topology" of the Rpt1 structure; however, Ref #29 discusses about BAF200 containing complex and its function in transcription of distinct genes, but nothing about Rpt1 structure. In fact, the PDB ID: 5GJK referred in the same paragraph is from Yan et al., JMB 2017. Did the authors mean to refer to Yan et al., JMB 2017?

- Page 7: Remove 's' from the word 'destabilizes' in "T214 is at the other end of INI1 α1 helix, facing the binding interface and mutation of this residue may destabilizes the helix as well as the binding interface (Supplemental Fig. S6D)."

- Page 13, line 5: Replace the word 'Comparison' with 'Comparision'

- Page 16, line 2: Add 's' to the sentence that starts as "Our study indicates...."

- Page 16, paragraph 2, line 7: Replace $>=500$ mM with ≥ 500 mM.
- Page 18: The sentence "Since Rpt1 domain is highly evolutionarily conserved, and since HIV-1 is of a recent origin, it is rather likely that TAR has evolved to mimic INI1 Rpt1 domain (instead of INI1 evolving to mimic TAR), understanding of which is likely to shed light on HIV-1 evolution." does not appear to be confusing. Isn't it that the IN:TAR interaction, which is critical for proper viral maturation pre-exists before IN encounters INI1 and then IN evolved to hijack and bind to INI1 using its TAR binding surface so that INI1 facilitate viral replication. However, if what the authors suggest is true, TAR evolved to counteract the effect of INI1 by protecting IN from binding to INI1, which is unlikely knowing that INI1 serves as dependency factor. Can the authors please provide more clarification on this issue why they thought it is the TAR that evolved but not IN?
- Page 24: About the GST-IN cloning. We all know that IN is a notorious protein to purify and maintain it soluble. However, the IN F185H solubility mutant has commonly been used for in vitro based assays in many Labs. When the authors say that they PCR out IN from HIV-1 HX3B clone, is it completely WT IN or did they also introduce the F185H solubility mutant in their clone? Please provide some details on this.
- Page 27: Please correct the font size of the "Biotin Labelled Seq..."
- Fig 3: In the body of the text of manuscript, panels E-G were referred but there is no panel labelled in Fig 3 nor in the caption. Please correct this.
- Fig 5: The label for panel A is cropped, please correct it. In addition to what has been discussed in the text, please provide the quantitative analysis of the morphological distribution of viral particles in a separate panel in this figure.
- Figure legends lack consistency. There are a lot of out of position periods (.) and please carefully proofread and correct all.
- Suppl. Fig S4 F: Please describe in the caption what the asterix (*) indicate on the blot shown.

Reviewer #2:

Remarks to the Author:

This manuscript by Bhutoria, Kalpana and colleagues presents the NMR structure of the HIV IN binding domain (RPT1) of INI1/SMARCB1 and computationally models binding of this domain to the IN CTD. The authors perform mutational analysis and in vitro binding studies to validate the modeled binding interface of INI1 RPT1 and IN CTD. The authors also find that IN CTD mutations that affect INI1 RPT1 binding also disrupt the interaction between IN CTD and HIV TAR RNA and that the INI1 RPT1 domain mimics (TAR) RNA structure, potentially contributing to its known diverse roles not only in various stages of the HIV life cycle but also in its cellular functions that include transcription and tumor suppression.

This is an important structure of the interaction domain of INI1, a critical cellular protein involved in various processes modeled to bind the HIV integrase, an interaction that led to the discovery of INI1 in 1994. The authors convincingly validate the INI1 RPT1 – IN CTD model using in vitro biochemical interaction studies and identify the critical residues that mediate the interaction by examining the effect of point mutants of INI-1 and IN CTD on the interaction.

Major comments:

1. The in vitro experiments validating the model should be fortified with more physiologically relevant data to demonstrate these proteins /TAR RNA actually bind in cells; co-immunoprecipitation experiments should be performed to show that INI1, IN, TAR-RNA are found in the same complex in

cells and that INI1 and TAR compete for IN binding. How much TAR RNA is pulled down with an IN immunoprecipitation experiment with increasing levels of INI1? Malignant Rhabdoid tumor cell lines that lack INI1 would be especially useful for these experiments enabling exogenous expression of increasing levels of not only wild-type and INI1 but also examination of the effect of INI1 point mutants defective for IN binding.

Related to this, the generation and functional examination of additional INI1 RPT1 point mutants, at residues determined by the modeling to be critical for IN interaction (such as those in Figure 6F) would be informative.

2. The effects of W235 IN mutants on viral replication, Gag processing, p24 production infectivity, etc (Figures 5B to H) need to be examined in context of the IN-INI1 interaction and clarified; It is unclear if the lack of INI1 binding in these mutants specifically causes these events, or if they are a consequence of the defect in integrase (with mechanisms independent of its binding to INI1). Knockdown studies of INI1/generation of virus in MRT lines and use of INI1 point mutants that according to the modeling would be defective for IN binding would be critical in characterizing these effects.

3. Does a W235 mutation in IN or INI1 mutant defective in IN binding prevent recruitment of INI1 into the viral particles? This can be addressed by collecting and concentrating Virus and examining presence of INI-1 by Western blotting.

4. Although the structure is of high importance, the model presented seems far-fetched and not supported with sufficient data. The functional roles for TAR-RNA binding by IN and/or competition with INI1 can have major implications on whether the vRNA is translated or encapsidated, resulting in defective viral assembly and defective cores. Experiments need to be performed to support the model, or the model needs to be modified with what is known i.e., that INI1 mimics TAR-RNA.

Minor comments:

1. Figures need to be numbered in the order in which they appear in the text. Moreover, all figures need to be described or alluded to in text.

2. On page 17, viral cDNA needs to be replaced with PIC (pre-integration complex).

Reviewer #3:

Remarks to the Author:

The manuscript by Bhutoria et al. present structure-function studies of the INI1-HIV integrase (IN) interaction. The authors present an NMR structure of a fragment of INI1 (residues 183-265) and a molecular model of a larger INI1 fragment (residues 183-304) with the latter being capable of binding to HIV IN. The authors then computationally docked a published CTD domain of IN onto their INI1183-304 model in order to elucidate possible key interactions, which were then used to create mutations in both INI1 and the IN-CTD for in vitro biochemical interaction studies. The authors then observed that many of the residues involved in the binding of IN to INI1 were important for the binding of IN to TAR RNA and competition experiments found that INI1 and TAR are capable of competing with each other for binding to the IN-CTD domain. Furthermore, IN mutants show similar patterns of binding to both TAR and INI1 in vitro and mutating W235 of the IN-CTD domain in vivo showed defects of capsid assembly in a manner similar to what was previously seen in the IN-TAR interaction and of integration. This then leads the authors to hypothesize that the interface between the IN-INI1 interaction mimics the interface between the IN-TAR interaction, which they follow up with by modeling the CTD-TAR complex and providing analysis of the resulting model. While the authors provide an interesting hypothesis and observation, a lot more work is needed on the functional side in order to establish the relevance their structure and models and strengthen the idea that there is structural mimicry between INI1 and TAR binding to IN-CTD. The work is certainly non rigorous enough to warrant publication as presented. Specific comments:

1) The authors do not provide any evidence to support the key interaction for their TAR-INI1 model. According to their model W235 interacts with G18 via hydrogen bonding. This binding mechanism

would be very unusual; tryptophans generally have stacking interactions with Adenines and Guanosines, and lysines make such proposed hydrogen bonding in protein-RNA interactions. This reviewer has no confidence in this key model. Titration of the protein into TAR would be required. And a bound structure to prove mimicry.

2) Mutations in TAR of loop residues (such G36 or C30 that are supposed to mirror D225 and E210 from INI1) and especially G18 that bind W235 of IN-CTD and testing their ability to bind the CTD are needed.

3) Binding curves for competition should include control RNAs to prove that the interaction is specific to TAR and not simply a non-specific charge-charge interaction.

4) The authors created mutations in the IN CTD that showed similar profiles of binding to both RPT1 and TAR (Fig. 4F-H). In order to strengthen their hypothesis that the two molecules are indeed structural mimics of each other, competition assays should be performed where key mutations in TAR, such as those listed in Fig. 6F, are unable to compete with the RPT1-CTD interaction (such as in Fig. 4D) and equivalent mutations in key INI1 residues are unable to compete with a TAR-CTD complex (such as in Fig. 4E).

5) The authors state previous work (ref. 30), which shows that INI1183-245 is sufficient for binding IN and that longer constructs enhance the binding affinity. However, the NMR structure of INI1183-265 solved by the authors that is then presumably used for modeling purposes does not bind IN through their assays (Supplemental Fig. S4B) whereas the larger construct that they use for modeling INI1183-304 does. The authors need data (perhaps an overlay of spectra) to show that the major features of the INI1183-265, especially W235 that is a major focus of the paper, are preserved in the INI1183-304 construct in order to provide confidence in their model of the latter. The authors should also propose a reason why their solved NMR construct does not match previous literature when it comes to binding IN?

6) The authors show that INI1183-304 can bind to both the CTD and the CCD of HIV IN (Supplementary Fig. 4D) and in Fig. 3B, they show that key mutations in their INI1-IN CTD model do not seem to significantly disrupt the binding of the CCD to INI1. The D225G and the T214A mutations do not completely abolish binding of the GST-IN to INI1183-304, suggesting there might be some compensatory mechanism or synergistic effect that the two domains have on each other. This might possibly affect some of the key structural interactions they have modeled in. Mutations in the CCD domain and testing the ability of the CTD to bind IN-CTD and IN can help to validate their NMR structure and model of the INI1183-304:IN-CTD interaction.

7) The authors show that mutations of a key interaction of the INI1-IN interaction, W235, in vivo result in very similar characteristics as mutations of virion morphogenesis seen in the IN-TAR interactions. Performing the same assays for mutations in other residues of their model, such as IN residues that interact with E210 and/or D225, would help to strengthen the idea that their INI1183-304:IN-CTD model is a molecular mimic of the TAR-IN interaction (residues R228 and/or R262 from their chart in Fig. 6F would be good candidates).

8) The line "Further analysis indicated a high degree of similarity between two desperate molecules" should say disparate.

9) The numbers for the INI1 construct in the following line "INI1183-304 NMR structure consists of a well-ordered region from aa 183-248 containing..." should be 183-265.

Response to reviewer's comments:

We thank the reviewers for their efforts and for many positive comments. We have addressed all the comments below.

Reviewer #1 (Remarks to the Author):

1. In Fig 1, the authors provide NMR structure of INI1 Rpt1 domain. Here, the authors should superimpose their structure and the crystal structure of INI1 Rpt1 domain and BAF155 SWIRM heterodimer [PDB ID: 5GJK; Yan et al JMB 2017]. Also, please check the NMR structure of Rpt1 [PDB ID: 5L7A/B; Sammak et al., FEBS 2018]) and discuss the similarities and the differences.

We have provided superimposition of all the known structures of Rpt1 domain of INI1 with our structure 6AX5 (**new Fig.1D**). All these structures demonstrate the presence of a well ordered Rpt1 domain(aa 183-248). The alignment of C-alpha atoms in this well-ordered Rpt1 (aa 183-248) region indicate that there is a nice overlap between all the structures with an RMSD of ~ 1:00 Å (**Supplemental Table ST2**). These structures differ from each other in the length, specifically in the linker region. The linker region is present in our structure, 6AX5 and in 5I7B (INI1₁₈₃₋₂₅₈) but not in others. In our structure it is from 249-265, and in 5I7B, it is from 249-258. Linker region in both these structures is disordered starting from aa 249.

A newly described CryoEM structure (PDBID: 6LTJ) is available (Feb 2020), which has been included in the above analysis. While both Rpt1 and Rpt2 regions are present in this CryoEM structure, the linker region is not present. We superimposed Rpt1 and Rpt2 regions of our modelled structure (Fig. 1E) individually with CryoEM structure (6LTJ). We found that Rpt1 and Rpt2 overlap nicely with an RMSD of ~1.0 and 1.25 Å respectively for the two domain (**Supplemental Table S3** for Rpt1, and data not shown for Rpt2). However, we found that flexibility of linker region positioned Rpt2 in two different spatial orientation in the modeled structure when compared to CryoEM structure. Since the linker and the Rpt2 are not involved in the interaction between INI1-Rpt1 and IN-CTD, and the linker is extremely flexible, the linker/Rpt2 conformation/orientation is unlikely to influence the INI1-Rpt1/IN-CTD complex structure. These information are included in the results and discussion sections.

2. In Fig 2, the authors provide a model of the IN CTD:INI1 Rpt1 interaction. HIV-1 IN exists in dimer↔tetramer equilibrium in solution. Can the authors discuss their model in this context, including the limitations of their model?

IN can exist in many multimeric states in solution, dimers, tetramers and even as octamers and dodecamers, as nicely illustrated by the CryoEM studies (Passos et al. Science, 2017). The minimal unit of an intasome (catalytically active form of IN bound to DNA) is a tetramer. The CryoEM structure indicates that within this tetramer, two monomers are close to the active site and the other two protomers are flanking the inner protomers. The arrangement of CTD from outer monomers of the four different monomers are not well resolved. However, it is interesting to note that while the domain structures within each monomer is the same, the linker that connects C-terminal domain (CTD) and the Central core domain (CCD) is flexible, allowing positioning of CTD in various special orientations (Fig. 3H of Passos et al). Interestingly, in these structures, while CTD from inner monomers are engaged in interaction with DNA the CTD outer protomers from higher order structure are not. It is likely that the INI1-Rpt1 can bind to CTD from the outer protomers and the INI1-Rpt1/IN-CTD complex structure will essentially remain the same as the one we proposed here, and binding of Rpt1 depends on whether CTD is free for interaction. It is interesting to note that the INI1 linker region (aa 146-265) is also flexible and it may allow the relative positioning of Rpt1 and Rpt2. It is conceivable that in a tetramer, while the CTD from inner

monomers are engaged in interaction with other domains of IN, the CTD from outer monomers may be engaged in binding to Rpt1 domain. Future CryoEM structures of the complex between IN/INI1 is likely to resolve these issues.

We have included a brief information about the issue regarding the flexibility of loops and how these proteins interact in multimeric state in the discussion starting from the sentence “The INI1 linker region (aa 146-265) is highly flexible”

Importantly, the authors should carefully evaluate the crystal structure reported by Yan et al., JMB 2017 and comment on the interaction surfaces overlap for HIV-1 IN and BAF155 SWIRM on INI1. Do IN and BAF155 SWIRM compete for INI1?

As suggested, we have superimposed and compared the IN-CTD/INI1-Rpt1 interaction to that of BAF155-SWRM/INI1-Rpt1 (PDB ID: 5GJK). BAF155-SWRM domain appears to bind to the same surface of INI1-Rpt1 as that of IN-CTD. However, it appears that INI1-Rpt1 binds with very extensive hydrogen bonding network with side chains of IN-CTD residues as compared to SWIRM domain, which shows some hydrogen bonding and also shows interaction with backbone atoms (Yan et al JMB 2017). This suggests that IN-CTD exhibits much broader and stronger interactions with Rpt1 domain than that of SWIRM domain. We surmise that it is possible that IN and BAF155 SWIRM domain compete with each other for binding to INI1. It is also likely that this interaction could displace INI1 from SWI/SNF complex. These are interesting experiments to do which are beyond the scope of this current report.

3. *In Suppl. Fig S4F, the authors showed in vitro pull-down data. The full-length IN binds significantly better than the two domains tested. This begs for additional quantitative data and the authors should also show:*

a. The interactions in cells by co-IP. Co-transfect INI1 and IN (WT or the different mutants tested in Fig 3) in the full-length context and show the binding efficiency.

As suggested, we have carried out co-immunoprecipitations of INI1 and IN (WT and different mutants) in the full-length context *in vivo*. The results are included in the **New Figure 5A**. We tested the interaction of WT IN and IN mutants, W235F, W235A, W235E, R228A and R269A/K273A for their ability to interact with WT INI1 by co-IP. Our model predicts that replacing tryptophan (W235) on IN with other non-polar or charged residues disrupts IN-INI1 interaction but not when replaced with another aromatic residue, phenylalanine. We are happy to report that the results of co-immunoprecipitation studies are consistent with our hypothesis. HA-INI1 was able to co-immunoprecipitate both WT IN and IN(W235F) *in vivo* but not the other mutants of IN.

b. The authors mentioned in the introduction that the stoichiometry of IN:INI1 in virions is ~1:2; however, it is not known whether that stoichiometry is a true reflection of the binary interaction in vivo. The authors should provide quantitative data using techniques such as analytical ultracentrifugation analysis to determine the stoichiometry of IN:INI1 complex.

Our attempts to determine the stoichiometry using biochemical means suggested that IN:INI1 forms higher order structures and multiple higher order complexes were observed. It is well known that IN can form dimers, tetramers, octamers and higher order structures in solution. INI1 also forms dimers, tetramers and octamers and higher order structures (ref: Das, Cano and Kalpana, JBC 2009). We believe that these factors played a role in biochemical assays we used (analytical ultracentrifugation) and complicated the interpretation of the data. We feel that future CryoEM studies are needed to resolve this issue and we are in the process of developing these assays to study IN-INI1 interactions.

4. *Results shown in Figs 3C, 4, and 5 primarily focus on the IN determinants of the IN:INI1 interaction. Others have shown similar phenotype using either ALLINIs/LEDGINs/NCINs or class II IN mutants. The class II IN mutants have effect on IN multimerization, the IN:TAR interaction,*

and hence perturb virion morphogenesis resulting in aberrant morphology (Gupta et al PLoS Biol 2016; Kessl et al., Cell 2016). Can the authors rule out whether the effect of the IN mutants described here is not due to IN multimerization defect?

There are several reports that indicate that some of the mutants we studied here are not defective for multimerization. Kessl et al., Cell 2016 demonstrated that the IN mutants K264A/K266A and R269A/K273A are not defective for multimerization *in vitro* but are defective for TAR RNA binding. In addition, Madison et al., (J. Virol. September 2017 Volume 91 Issue 17 e00821-17) proposed that the defect in infectivity of virion particles harboring R269A/K273A are not due to defect in multimerization of IN but it is due to their inability to bind to RNA *in vivo*. These studies indicated that while some (but not all) class II IN mutants show aberrant multimerization, this property alone is not sufficient to describe the defect of these mutants in RNA binding or IN1 interaction.

Another mutant we studied here is W235E, which is considered as a class I mutant (Cannon et al, J.Virol, 1996; Engelman et al, Adv Virus Res, 1999; Li et al, JBC, 2005). This mutant is not defective for binding to LEDGF (current data, See Fig. 6) and is not defective for single end *in vitro* integration activity (Cannon et al, J.Virol, 1996; Li et al, JBC, 2005), suggesting that this mutant exhibits normal multimerization. The selective defect of W235E mutant to bind to IN1 and TAR RNA (Fig. 3 and 4) indicate that the defect we observe *in vivo* is likely not due to defect in multimerization. We would like to point out that thus far, all IN mutants that are defective for RNA binding were also defective for binding to IN1, consistent with the modeling data and *in vitro* binding data.

5. The experimental results shown in Fig 5 are tricky to understand. Based on the data shown in Fig 5A and several previous reports, these defective viruses are expected to be completely dead for completion of reverse transcription; however, the authors showed that the defect is only at the integration step (Fig 5E-H). Can they comment on this discrepancy?

Please note that the **Figure 5 is now New Figure 6**. First, the data that W235E IN mutant is not defective for reverse transcription and is selectively defective for integration is consistent with the previously published studies (Cannon et. al. J. VIROLOGY, Jan. 1996, p. 651–657 Vol. 70). Here we confirmed this previous data using qPCR-based assays and further demonstrated that this mutant is selectively defective for binding to IN1 and TAR RNA. Furthermore, Cannon et al. concluded that the W235E mutant does not exhibit morphological defects based on EM analysis. But close examination of the images published in their manuscript (Cannon et al. Fig. 4G) clearly shows the presence of defective particles. Specifically, one can see the presence of empty cylindrical cores with electron-dense material outside the cores. Therefore, our results are consistent with the published results.

Second, W235E exhibited the presence of a mixture of conical, non-conical and dual or nested cores (**Figure 6B**). We have now included the percentages of the various morphologies present in W235E mutant in supplemental material (**Supplemental Figure S8B**). It appears that ~ 32% (23.1+ 1.5 +6.9, see **Fig. S8B**) of all the cores exhibited normal electron dense condensate. While this number is much lower than that compared to wild type (70%), presence of these capsids with normal condensate may have accounted for the lack of defect in reverse transcription in case of the W235E mutant. Based on our results, we speculate that the disruption in the capsid morphology is milder in case of W235E mutant and that not all defects in particle morphology leads to defect in reverse transcription.

Third, W235E mutant is considered as class I mutant as it is not defective for integration *in vitro*, indicating that it is catalytically functional but is defective for integration *in vivo* and does not fit the classical definition of a class II mutant. This residue is at the interface of IN-CTD/INI1-Rpt1 complex. Results of both *in vitro* and *in vivo* binding studies are consistent with the hypothesis that this residue is important for IN/INI1 interaction (**New Figs. 5A**) and now we also show that this mutant does not incorporate IN1 into virions (**New Fig. 6B**).

Minor Comments: - Page 4: Replace the PDB ID: 5GJK with your own structure PDB ID: 6AX5. Thanks for pointing out this. We have now changed it.

- Ref #29 was cited for the “ $\beta\beta\alpha$ topology” of the Rpt1 structure; however, Ref #29 discusses about BAF200 containing complex and its function in transcription of distinct genes, but nothing about Rpt1 structure. In fact, the PDB ID: 5GJK referred in the same paragraph is from Yan et al., JMB 2017. Did the authors mean to refer to Yan et al., JMB 2017?

We have corrected this – Thank you for pointing this out.

- Page 7: Remove ‘s’ from the word ‘destabilizes’ in “T214 is at the other end of INI1 α 1 helix, facing the binding interface and mutation of this residue may destabilizes the helix as well as the binding interface (Supplemental Fig. S6D).”

Done.

- Page 13, line 5: Replace the word ‘Comparation’ with ‘Comparison’.

Done.

- Page 16, line 2: Add ‘s’ to the sentence that starts as “Our study indicates....”

Done.

- Page 16, paragraph 2, line 7: Replace ≥ 500 mM with ≥ 500 mM.

Done.

- Page 18: The sentence “Since Rpt1 domain is highly evolutionarily conserved, and since HIV-1 is of a recent origin, it is rather likely that TAR has evolved to mimic INI1 Rpt1 domain (instead of INI1 evolving to mimic TAR), understanding of which is likely to shed light on HIV-1 evolution.” does not appear to be confusing. Isn’t it that the IN:TAR interaction, which is critical for proper viral maturation pre-exists before IN encounters INI1 and then IN evolved to hijack and bind to INI1 using its TAR binding surface so that INI1 facilitate viral replication. However, if what the authors suggest is true, TAR evolved to counteract the effect of INI1 by protecting IN from binding to INI1, which is unlikely knowing that INI1 serves as dependency factor. Can the authors please provide more clarification on this issue why they thought it is the TAR that evolved but not IN?

The reviewer provides an interesting hypothesis. However, based on all the observations, it is likely that all three, i.e. INI1, RNA and GagPol exists in the cytoplasm during assembly within the cells. INI1 binds to IN within GagPol and may act as a “place-holder” to prevent untimely binding of RNA to IN during assembly and until the time of maturation. This “place-holder” function has been noted for other RNA mimicking proteins (Please see the discussion). Please see the discussion starting from “**Significance of INI1 Rpt1 RNA mimicry to HIV-1 replication:**”

With regard to cellular function of INI1, we propose that it has evolved to mimic a cellular RNA, the 7SK RNA. Please see the discussion starting from “**INI1 RNA mimicry and its significance to viral and cellular transcription**”.

- Page 24: About the GST-IN cloning. We all know that IN is a notorious protein to purify and maintain it soluble. However, the IN F185H solubility mutant has commonly been used for in vitro based assays in many Labs. When the authors say that they PCR out IN from HIV-1 HX3B clone, is it completely WT IN or did they also introduce the F185H solubility mutant in their clone? Please provide some details on this.

We have used IN-CTD in most of our in vitro studies. For studies with IN-CTD, F185H substitution does not matter, as this residue is not in the C-terminal domain. CTD is ~200-228 aa. It is true that IN is insoluble when it is highly concentrated. But at lower concentrations, it is soluble with relatively high salt. F185H mutation is necessary when one needs to purify it to high concentrations for structural studies. We have used full length wild type IN for some of the *in vitro* studies. But for these studies, we do not need very high concentration of IN.

- Page 27: Please correct the font size of the “Biotin Labelled Seq...”

Done. These were in different Font but the same size.

- Fig 3: In the body of the text of manuscript, panels E-G were referred but there is no panel labelled in Fig 3 nor in the caption. Please correct this.

Thanks for noting this. This is taken care of.

- Fig 5: The label for panel A is cropped, please correct it. In addition to what has been discussed in the text, please provide the quantitative analysis of the morphological distribution of viral particles in a separate panel in this figure.

We have provided the morphological analysis in a separate table in the supplement. Please see supplemental **Figure S8B**.

- Figure legends lack consistency. There are a lot of out of position periods (.) and please carefully proofread and correct all.

Thank you for the information. This is taken care of.

- Suppl. Fig S4 F: Please describe in the caption what the asterix (*) indicate on the blot shown. Done.

Reviewer #2 (Remarks to the Author):

Major comments:

1. *The in vitro experiments validating the model should be fortified with more physiologically relevant data to demonstrate these proteins/TAR RNA actually bind in cells; co-immunoprecipitation experiments should be performed to show that INI1, IN, TAR-RNA are found in the same complex in cells and that INI1 and TAR compete for IN binding. How much TAR RNA is pulled down with an IN immunoprecipitation experiment with increasing levels of INI1? Malignant Rhabdoid tumor cell lines that lack INI1 would be especially useful for these experiments enabling exogenous expression of increasing levels of not only wild-type and INI1 but also examination of the effect of INI1 point mutants defective for IN binding. Related to this, the generation and functional examination of additional INI1 RPT1 point mutants, at residues determined by the modeling to be critical for IN interaction (such as those in Figure 6F) would be informative.*

We have carried out experiments to address reviewer's comments by conducting experiments *in vivo* and we are happy to note that the results of these experiments are consistent with our hypothesis. We have split up the reviewer's question into three parts (a, b and c).

1a. *Are INI1, IN and TAR-RNA found in the same complex in MON and if they do, what is the effect of increasing concentration of INI1?*

As suggested by the reviewer, we carried out experiments in MON and found that pulling down IN results in co-immunoprecipitation of INI1 as well as TAR RNA, indicating that these three components are present in the same complex (**See new Fig 5B**). In addition, we show that addition of increasing concentration of INI1 decreases the amount of TAR RNA bound by IN in a dose dependent manner (**See Fig. 5B**). These results are consistent with our hypothesis and are also consistent with the *in vitro* competition experiment.

1b. *Does the IN-binding defective INI1 mutant compete with TAR RNA for binding to IN in vivo?*

We have carried out the RNA-co-IP experiment to determine if IN-binding-defective mutant compete with TAR RNA *in vivo*. The answer to this is "no", consistent with our hypothesis. We carried out RNA-co-IP experiment *in vivo* using INI1 mutants D225G and

D225E. The substitution of Aspartate with Glycine alters the charge on INI1 and substitution with glutamate retains that charge. The results of these experiments indicate that INI1(D225G) does not bind to IN and is unable to compete with TAR to bind to IN *in vivo*. On the contrary, INI1(D225E) binds to IN and also competes with TAR similar to wild type INI1 (**New Fig. 5C**). Thus, these results establish that INI1 binding to IN is necessary to compete with TAR RNA to bind to IN *in vivo*.

1c. Functional examination of INI1 mutant:

Our hypothesis is that INI1 protein binds to IN within GagPol and facilitates particle production. We have reported earlier that transfection of MON cells with viral vectors show very little particle production and addition of INI1 complements this defect and increases particle production (Sorin et al, *Retrovirology*, 2006). Here we tested D225G, which is defective for binding to IN and mutant D225E that has conservative substitution to see their effect on particle production. We find that INI1 increases particle production, INI1(D225G) mutant does not and INI1(D225E) mutant does (**New Fig. 5D**). These results are consistent with our hypothesis and phenotype of mutants that are defective for binding to IN.

2. The effects of W235 IN mutants on viral replication, Gag processing, p24 production, infectivity, etc (Figures 5B to H) need to be examined in context of the IN-INI1 interaction and clarified; It is unclear if the lack of INI1 binding in these mutants specifically causes these events, or if they are a consequence of the defect in integrase (with mechanisms independent of its binding to INI1). Knockdown studies of INI1/generation of virus in MRT lines and use of INI1 point mutants that according to the modeling would be defective for IN binding would be critical in characterizing these effects.

To explain the function of INI1 and the phenotype of IN mutants defective for binding to INI1 or RNA, we propose a model that is presented in detail in the discussion section (**New Figure 8E**). Because RNA and INI1 share the same binding surface on IN, it is difficult to distinguish between the effects from these two molecules on particle morphogenesis. However, we propose that both INI1 and RNA binding to IN are necessary but in different space and time. Please see the discussion starting from the paragraph “What might be the role of TAR RNA mimicry by INI1-Rpt1 in HIV-1 replication?”

We suggest that INI1 binds to GagPol to prevent steric hindrance caused by the untimely binding of RNA during assembly. Thus INI1 functions as a “place-holder” by binding to RNA-binding domain of IN within GagPol during assembly and prevents RNA from binding to it till the time of particle morphogenesis. But INI1 is not be required if RNA is not binding to IN. Thus, IN mutants that are defective for binding to both INI1 and RNA assemble and the particle exit the cells. However, during morphogenesis RNA (or INI1 binding) is necessary and hence the IN mutants get defective morphogenesis. At this stage we cannot distinguish between the requirement of RNA versus INI1 for morphogenesis. All the known evidence support this model, where knocking-down INI, INI1^{-/-} cell lines or expression of a dominant negative mutant of INI1 all lead to disruption of assembly and particle production of the wild type virus. However, IN mutants are not defective for particle production.

We tested the function of INI1 mutants D225G and D225E on their ability to support particle production in INI1^{-/-} MON cells. INI1(D225G) is defective for binding to IN (**New Fig. 5C**) and hence it should not support particle production. However, D225E is able to bind to IN (**New Fig. 5C**) and it should support particle production. Our analysis of viral particle production in MON cells in the presence and absence of INI1 and mutants are consistent with this hypothesis (**New Fig. 5D**).

3. Does a W235 mutation in IN or INI1 mutant defective in IN binding prevent recruitment of INI1

into the viral particles? This can be addressed by collecting and concentrating Virus and examining presence of INI-1 by Western blotting.

As requested by the reviewer, we have produced and concentrated the virions through sucrose cushion and subjected these virions for Western analysis using anti-INI1, anti-IN and anti-p24 antibodies. Our data very nicely illustrates that INI1/hSNF5 is present in wild type viruses but its incorporation is drastically diminished in the IN mutants defective for binding to INI1 (W235E and R228A). We also demonstrate that all these virions incorporate equal amount of IN and mutants. This defect is also not due to the lack of/reduced INI1 in the cells transfected with mutant viruses. These data confirm our previous findings that INI1 is incorporated into virions and provide additional information that INI1-interaction defective IN mutants are impaired for incorporation of INI1 into the virions. These data are included in the **Figure 6C**.

4. *Although the structure is of high importance, the model presented seems far-fetched and not supported with sufficient data. The functional roles for TAR-RNA binding by IN and/or competition with INI1 can have major implications on whether the vRNA is translated or encapsidated, resulting in defective viral assembly and defective cores. Experiments need to be performed to support the model, or the model needs to be modified with what is known i.e., that INI1 mimics TAR-RNA.*

To address the reviewer's comment, we have used available evidence and simplified the model to illustrate the fact that INI1 and TAR compete for binding to IN within the producer cells and to illustrate its consequence on assembly and particle production. This model is presented in new **Fig. 8E**.

Minor comments: 1. *Figures need to be numbered in the order in which they appear in the text. Moreover, all figures need to be described or alluded to in text.* This is Done. 2. *On page 17, viral cDNA needs to be replaced with PIC (pre-integration complex).* Please note that the introduction is reduced to fit the word limit and we have changed the sentences.

Reviewer #3 (Remarks to the Author):

The manuscript by Bhutoria et al. present structure-function studies of the INI1-HIV integrase (IN) interaction. While the authors provide an interesting hypothesis and observation, a lot more work is needed on the functional side in order to establish the relevance their structure and models and strengthen the idea that there is structural mimicry between INI1 and TAR binding to IN-CTD. The work is certainly non rigorous enough to warrant publication as presented.

We are really surprised about the above comment and we respectfully disagree with the reviewer that the work is not rigorous enough. We request the reviewer to carefully consider the manuscript and the responses provided below. The similarity in Rpt1 domain of INI1 and TAR RNA was arrived from comparing the individual NMR structures and docking of CTD to TAR RNA provided the supporting evidence for our hypothesis. But we came to the hypothesis by conducting an extensive series of experiments, that includes the following:

i) generation of NMR structure of INI1-Rpt1; 2) modeling INI1-Rpt1/IN-CTD interaction; iii) validating the model by mutational studies *in vitro* and *in vivo*; iv) demonstrating the competition between INI1-Rpt1 and TAR RNA to bind to IN *in vitro* and *in vivo*; v) functional virological studies to demonstrate that IN mutants are defective for incorporation of INI1 into virions, defect of virions for virion morphology; vi) functional studies to demonstrate that INI1 is required for assembly and particle production *in vivo* and that wild type INI1 but not IN-binding-defective INI1 mutants are not able to complement this defect; vii) modeling (and repeating it again this time) of TAR RNA binding to IN-CTD, which shows complete superposition of TAR and INI1-Rpt1 that support our

hypothesis; and finally viii) comparison of surface charge distribution of INI1-Rpt1 and TAR RNA NMR structures demonstrating the mimicry.

We also point out that many of the experiments requested by the reviewer is already performed or they are already reported in the literature. We have indicated the details of these in the response below.

Specific comments:

1) The authors do not provide any evidence to support the key interaction for their TAR-INI1 model. According to their model W235 interacts with G18 via hydrogen bonding. This binding mechanism would be very unusual; tryptophans generally have stacking interactions with Adenines and Guanosines, and lysines make such proposed hydrogen bonding in protein-RNA interactions. This reviewer has no confidence in this key model. Titration of the protein into TAR would be required. And a bound structure to prove mimicry.

First, we are not presenting TAR-INI1 model, but TAR/IN-CTD complex model. There is no W235 residue in either INI1-Rpt1 or TAR. Second, we have extensive interaction and functional studies to show that the same IN residues are involved in binding to INI1-Rpt1 and RNA, which is borne out by this RNA/IN interaction model.

To address reviewers concern, we have now collaborated with computational modelling experts and have redone both INI1₁₈₃₋₃₀₄/IN-CTD and TAR/IN-CTD models (**Fig. 7**), using another docking program (MDockPP) that is more systematic in searching the conformational space than HADDOCK. Analysis of these new modeled complex structures reveal that the binding resulted from specific interactions between the lysine and arginine residues of IN-CTD and the backbones of TAR. The newly modeled complex TAR/CTD structure features the same binding interface of CTD as in the Rpt1/CTD complex. This finding strongly supports our conclusion of Rpt1 being a structural mimic of TAR. We find that the residue W235 is not contributing to the binding through any ionic interactions and its interaction is non-ionic. We also examined different local conformations of the side-chain of W235, and concluded that there is no possible way for the residue to have stacking interaction with any RNA bases in this binding pose. These new analyses have been included in the text (**See new Figure 7**).

2) Mutations in TAR of loop residues (such G36 or C30 that are supposed to mirror D225 and E210 from INI1) and especially G18 that bind W235 of IN-CTD and testing their ability to bind the CTD are needed.

The reviewer is asking us to substitute the G36 and C30 residues to test if they are necessary for binding to IN. Kessl et al. already investigated the TAR region that is required for binding to IN (Kessl et al Cell 2016). i) They carried out CLIP-seq analysis and found that residue at position 30 (C30) of TAR sequence was the most preferred residue for binding to IN; ii) They also deleted the loop and the bulge region of the TAR RNA and found that deletion of these structures reduced the binding to IN but substitution of the residues in the loop and bulge did not compromise binding. We believe that this is because negatively charged phosphate groups of the nucleotides in the bulge are contacting IN and hence substituting these nucleotides are not likely to affect binding. These studies are in agreement with our proposed model.

3) Binding curves for competition should include control RNAs to prove that the interaction is specific to TAR and not simple a non-specific charge-charge interaction.

The interaction between TAR and IN are well characterized by Kessl et. al. They tested the interaction of various segments of viral RNA and non-specific RNAs and determined the K_d values for each using the same Alpha assay. They found that TAR has the lowest K_d value.

Furthermore, we have used region of viral RNA other than TAR and found that it is unable to inhibit IN-CTD/INI1-Rpt1 interaction in our competition studies (**please see Supplemental Figure S7B**). These studies establish that binding of IN to TAR is not a simple charge-charge interaction.

4) The authors created mutations in the IN CTD that showed similar profiles of binding to both RPT1 and TAR (Fig. 4F-H). In order to strengthen their hypothesis that the two molecules are indeed structural mimics of each other, competition assays should be performed where key mutations in TAR, such as those listed in Fig. 6F, are unable to compete with the RPT1-CTD interaction (such as in Fig. 4D) and equivalent mutations in key INI1 residues are unable to compete with a TAR-CTD complex (such as in Fig. 4E).

We have already shown that TAR and Rpt1 compete with each other for binding to IN-CTD *in vitro* (**Fig. 4D and E**). Now we have demonstrated that TAR and INI1 competes with each other to bind to IN *in vivo* (**Fig. 5B**) Furthermore, we have demonstrated that non-binding mutant INI1(D225G) is unable to compete with TAR to bind to IN (**Fig. 5C**). The results of these experiments are consistent with our hypothesis and addresses reviewer's concern.

5) The authors state previous work (ref. 30), which shows that INI1183-245 is sufficient for binding IN and that longer constructs enhance the binding affinity. However, the NMR structure of INI1183-265 solved by the authors that is then presumably used for modeling purposes does not bind IN through their assays (Supplemental Fig. S4B) whereas the larger construct that they use for modeling INI1183-304 does. The authors need data (perhaps an overlay of spectra) to show that the major features of the INI1183-265, especially W235 that is a major focus of the paper, are preserved in the INI1183-304 construct in order to provide confidence in their model of the latter. The authors should also propose a reason why their solved NMR construct does not match previous literature when it comes to binding IN?

First, the reviewer is confused as to where the W235 residue is. This residue is not on INI1 but it is on IN. Therefore, we cannot identify W235 on INI1 NMR. Second, the model of INI1₁₈₃₋₃₀₄ was derived from the NMR structure of INI1₁₈₃₋₂₆₅ and hence the aa 183-265 regions are identical in the NMR structure and the model. Now we provide superimposition of our NMR structure with the model and also provide RMSD to show that they are all in agreement (**Fig.1D and Supplemental Table ST3**). Therefore, the Rpt1 portion is identical in INI1₁₈₃₋₂₆₅ NMR and in INI1₁₈₃₋₃₀₄ modeled structures.

6) The authors show that INI1183-304 can bind to both the CTD and the CCD of HIV IN (Supplementary Fig. 4D) and in Fig. 3B, they show that key mutations in their INI1-IN CTD model do not seem to significantly disrupt the binding of the CCD to INI1. The D225G and the T214A mutations do not completely abolish binding of the GST-IN to INI1183-304, suggesting there might be some compensatory mechanism or synergistic effect that the two domains have on each other. This might possibly affect some of the key structural interactions they have modeled in. Mutations in the CCD domain and testing the ability of the CTD to bind IN-CTD and IN can help to validate their NMR structure and model of the INI1183-304:IN-CTD interaction.

The reviewer is asking how the CCD portion of IN influence CTD binding to INI1. In the revised manuscript we have created the mutations in full-length IN and tested the interaction of WT and mutant INs with full length INI1 by co-immunoprecipitations (**New Fig. 5A**). Furthermore, we have tested the ability of INI1 to compete with RNA for binding to IN *in vivo* (**New Fig. 5B**). We find that: i) The CTD mutations in the context of full-length IN are unable to bind to full-length INI1 *in vivo* except for W235F; (ii) while wild type INI1 is able to compete with RNA to bind to wild type IN, the mutant INI1(D225G) is unable to compete with RNA *in vivo* (**New Fig. 5C**). Furthermore, W235E and R228A mutations in the context of full length IN do not incorporate INI1

into the virions (**Fig. 6C**). These studies validate our model in the full-length context and *in vivo*, strengthening our model.

7) *The authors show that mutations of a key interaction of the INI1-IN interaction, W235, in vivo result in very similar characteristics as mutations of virion morphogenesis seen in the IN-TAR interactions. Performing the same assays for mutations in other residues of their model, such as IN residues that interact with E210 and/or D225, would help to strengthen the idea that their INI1183-304:IN-CTD model is a molecular mimic of the TAR-IN interaction (residues R228 and/or R262 from their chart in Fig. 6F would be good candidates).*

Based on our model, residues R228 interacts with INI1-D225, and residues K244 and R262 interact with INI1-E210, respectively. It has been reported that R228A and R262A mutations lead to production of defective particles (Lu et al. *J Virol* **79**, 10356, 2005). We provide the additional evidence that: IN mutants R228A (that interact with INI1-D225) and K244A (that interact with INI1-E210) disrupt the interaction of IN-CTD with INI1₁₈₃₋₃₀₄ (**Fig. 4G**); W235E and R228A mutations disrupt interaction of IN with INI1 *in vivo* (**new Fig. 5A**); and W235E and R228A mutants makes the virion defective for incorporation of INI1 (**new Fig. 6C**). These data are consistent with our model and addresses reviewers' comment.

8) *The line "Further analysis indicated a high degree of similarity between two desperate molecules" should say disparate.*

Thanks you for pointing this out. We have corrected this.

9) *The numbers for the INI1 construct in the following line "INI1183-304 NMR structure consists of a well-ordered region from aa 183-248 containing..." should be 183-265.*

We have corrected this.

Reviewers' Comments:

Reviewer #2:

Remarks to the Author:

In their revised manuscript, Kalpana and colleagues provide compelling evidence for structural mimicry between the INI-1 Rpt1 domain and HIV-1 TAR RNA and present a model in which binding of INI-1 or TAR to the HIV-1 IN during HIV-1 assembly facilitates replication.

My main concerns to the original version of the manuscript itemized previously had to do with the potential physiological relevance of the described interactions in context of an infected cell (also shared by the first reviewer), in particle formation and on viral replication. The authors have made an extensive effort and experimentally addressed all of my concerns. The authors have performed a number of co-transfection and co-immunoprecipitation experiments, characterizing IN binding by INI-1 and TAR and using mutant IN and INI-1 RPT1 mutants. Inclusion of this additional data has led to a more refined simplified model for functional consequences of the competitive binding of TAR and INI-1 to IN.

Reviewer #3:

Remarks to the Author:

The revised manuscript is significantly improved with the help of new structural data to aid in modeling interactions of the HIV-1 IN-CTD domain with INI1-Rpt1 and TAR coupled with increased biochemical and in vivo studies conducted to validate the INI1-IN interaction and the competition between INI1 and TAR for the binding of HIV-1 IN-CTD. The manuscript is suitable for publication perhaps after the following comments are addressed:

1) The modeling of the IN-CTD/TAR complex is strongly supported from the viewpoint of the IN-CTD interacting domain by both in vivo and in vitro experiments as to which residues are critical for this binding event. However, as the molecular modeling of key interactions changed with the use of a different docking program, it would be good to validate the model from the RNA perspective to give credit to their modeling, especially given that arginines such as R228 and R263 would intuitively be capable of interacting with the TAR bulge capable that is capable of forming an arginine sandwich motif on the other side of the helix from what is currently modeled. Performing assays with mutations of key residues of TAR from their model to assess the effects on the K_d of the IN-CTD/TAR interaction and the ability of such TAR mutants to compete with the INI1/IN-CTD interaction would significantly improve confidence that the interactions are structural mimics.

2) As TAR is a molecule with multiple dynamic states, especially in the modeled region where the authors propose the IN-CTD binds, without a structure of this complex it would be difficult to know whether this interaction would be a true structural mimetic to the INI1/IN-CTD complex. Toning down the definitive nature of the mimicry would be prudent in this situation, especially without any validation of the complex. In particular, the mechanisms by which the tryptophan, lysines, and arginines interact with the TAR RNA in the revised model are highly unusual and warrant some discussion of why this model would be a feasible possibility.

Point-by-point responses to reviewers' comments

We appreciate the efforts of the reviewers in reviewing this manuscript and are pleased by the positive comments. Please see the point-by-point response to the reviewer's comments below.

Reviewer #2 (Remarks to the Author):

In their revised manuscript, Kalpana and colleagues provide compelling evidence for structural mimicry between the INI-1 Rpt1 domain and HIV-1 TAR RNA and present a model in which binding of INI-1 or TAR to the HIV-1 IN during HIV-1 assembly facilitates replication. My main concerns to the original version of the manuscript itemized previously had to do with the potential physiological relevance of the described interactions in context of an infected cell (also shared by the first reviewer), in particle formation and on viral replication. The authors have made an extensive effort and experimentally addressed all of my concerns. The authors have performed a number of co-transfection and co-immunoprecipitation experiments, characterizing IN binding by INI-1 and TAR and using mutant IN and INI-1 RPT1 mutants. Inclusion of this additional data has led to a more refined simplified model for functional consequences of the competitive binding of TAR and INI-1 to IN.

We appreciate the positive comments of the reviewer and we are delighted that our revisions were satisfactory.

Reviewer #3 (Remarks to the Author):

The revised manuscript is significantly improved with the help of new structural data to aid in modeling interactions of the HIV-1 IN-CTD domain with INI1-Rpt1 and TAR coupled with increased biochemical and in vivo studies conducted to validate the INI1-IN interaction and the competition between INI1 and TAR for the binding of HIV-1 IN-CTD. The manuscript is suitable for publication perhaps after the following comments are addressed:

We appreciate the comments of the reviewer and the recognition that the INI1-Rpt1 and TAR interactions are strengthened by the new data.

1) The modeling of the IN-CTD/TAR complex is strongly supported from the viewpoint of the IN-CTD interacting domain by both in vivo and in vitro experiments as to which residues are critical for this binding event. However, as the molecular modeling of key interactions changed with the use of a different docking program, it would be good to validate the model from the RNA perspective to give credit to their modeling, especially given that arginines such as R228 and R263 would intuitively be capable of interacting with the TAR bulge capable that is capable of forming an arginine sandwich motif on the other side of the helix from what is currently modeled. Performing assays with mutations of key residues of TAR from their model to assess the effects on the Kd of the IN-CTD/TAR interaction and the ability of such TAR mutants to compete with the INI1/IN-CTD interaction would significantly improve confidence that the interactions are structural mimics.

We respectfully point out that all the molecular interactions essentially remained the same by using the two different docking programs HADDOCK and MDockPP. The difference in the two docking program used lies in the kind of constraints we used for screening the complex poses. When we docked using HADDOCK, the docking was done *ab initio*, without providing any interaction constraints as we did not know the residues that were important for interactions at that time. However, when we docked it a second time using MDockPP, we included the constraints based on the *in vitro* binding data between IN-CTD and INI1-Rpt1 interactions. We were pleased

to see that in both the docking programs, the interface residues interactions remained the same, providing further support to our docking models.

The reviewer points out that “*arginines such as R228 and R263 would intuitively be capable of interacting with the TAR bulge that is capable of forming an arginine sandwich motif on the other side of the helix from what is currently modeled*” and that we need to carry out mutagenesis of TAR to further characterize the interaction.

Based on our modelling and based on the previously reported mutagenesis data, we do not find the evidence for the arginine sandwich between IN and TAR RNA. Previous mutagenesis data was reported by the Kessel et al (Cell 166, 1257–1268, August 25, 2016, <http://dx.doi.org/10.1016/j.cell.2016.07.044>). In that report, they found that deletion of the loop, as well as the 3 nucleotide bulge of TAR RNA severely reduces the ability of TAR RNA to bind to IN indicating that loop and bulge structures are important for TAR RNA binding. They also created substitution mutations of loop and bulge structure, without altering the length or shape of these structures. They created “TAR(m22-24,m29-34)” construct, which had the bulge (nucleotides 22-24) and loop (nucleotides 29-34) sections mutated to AGA (from UCU) and GAGAGA (from CUGGA) respectively; and “TAR(m29-34)” construct, which had the 29-34 section mutated to CUCUCU. They found that both of these substitutions in the loop and 3-nt bulge did not significantly compromise IN binding to vRNA(1–57)-TAR (Supplementary Figure S3c).

The above results indicate that IN recognizes the loop and bulge structures of TAR. However, it does not appear to differentiate the sequences in the bulge and loop regions. These results strongly imply that specific base interactions may not be occurring between IN residues and the nucleotides of TAR RNA. These results are consistent with our proposal that the positively charged residues of IN are recognizing the spatial positioning of negatively charged phosphate groups on TAR RNA as its binding is independent of base sequence. These results makes a strong argument that TAR-IN interactions may not use the arginine-sandwich binding mode.

Additional information from the Kessel et al report is that they found that nucleotide 30 (C30) located in the loop was essential for interaction of TAR RNA with IN, based on the CLIP-seq data (refer to Figures 1C, S2C and S3 in Kessel et al report). This data also supports our model as our data indicates that C30 residue of TAR is involved in binding to IN.

Three dimensional structural determination of TAR RNA bound to IN is likely to provide detailed mechanism of this interaction. However, this is beyond the scope of this article.

2) As TAR is a molecule with multiple dynamic states, especially in the modeled region where the authors propose the IN-CTD binds, without a structure of this complex it would be difficult to know whether this interaction would be a true structural mimetic to the IN1/IN-CTD complex. Toning down the definitive nature of the mimicry would be prudent in this situation, especially without any validation of the complex. In particular, the mechanisms by which the tryptophan, lysines, and arginines interact with the TAR RNA in the revised model are highly unusual and warrant some discussion of why this model would be a feasible possibility.

We respectfully point out that while the concept of mimicry was uncovered based on our observation of binding patterns of IN mutants, the similarity in two structures were noted even in the unbound NMR structure of the two molecules. Our observation was that the binding pattern of IN mutants to either TAR RNA or IN1 Rpt1 were identical, and that TAR RNA and IN1 Rpt1 competed with each other with identical IC50 values. This observation prompted us to investigate the similarity in the two NMR structures of the TAR RNA (PDB ID: 1ANR) and IN1-Rpt1 (PDB ID: 6AX5). We found that the three dimensional surface electrostatic properties of the NMR structure of IN1-Rpt1 shows an arrangement of negative charges that was remarkably similar to the spatial arrangement of phosphate groups of the TAR RNA as seen in the Figure 8a. Furthermore, when the IN1-Rpt1/IN and TAR/IN complexes were superimposed, the negatively charged interface

residues of IN11 were in close proximity to the phosphate groups of TAR (Supplemental Figure 9b). All these results together prompted us to suggest RNA mimicry.

But nevertheless, as suggested by the reviewer, we have toned down the information on RNA mimicry and have used “suggested” “possible” RNA mimicry in the manuscript. Furthermore, we have included the following results from the previous report in the manuscript under results section.

Last paragraph, Page 15 (highlighted):

“Previous report indicated that nucleotide C30 of TAR and the loop were important for IN binding using CLIP-seq assay¹⁸. Furthermore, deletions, but not the nucleotide substitutions, of the loop and the adjacent three nucleotide bulge of TAR reduced the IN binding, suggesting that IN prefers a specific structural element of TAR rather than a specific nucleotide sequence¹⁸. Our model is consistent with this report and suggests that spatial positioning of the phosphate groups from the loop and the bulge are important for interaction with IN residues.”